



# Controls on the distribution of cosmogenic [10]Be across shore platforms

Martin D. Hurst[1,2], Dylan H. Rood[3], and Michael A. Ellis[2]

[1]School of Geographical and Earth Sciences, University of Glasgow, Glasgow, G12 8QQ, UK.
[2]British Geological Survey, Nicker Hill, Keyworth, Nottingham, NG12 5GG, UK.
[3]Department of Earth Science and Engineering, Imperial College London, South Kensington Campus, London SW7 2AZ, UK

*Correspondence to:* Martin D. Hurst (Martin.Hurst@glasgow.ac.uk)

**Abstract.** Quantifying rates of erosion on cliffed coasts across a range of timescales is vital for understanding the drivers and processes of coastal change and for assessing risks posed by future cliff retreat. Historical records cover at best the last 150 years; Cosmogenic radionuclides, such as [10]Be could allow us to look further into past to assess coastal change at millenial timescales. CRNs accumulate *in-situ* near the Earth surface and have been used extensively to quantify erosion rates, burial
dates and surface exposure ages in terrestrial landscapes over the last three decades. More recently, applications in rocky coast settings have quantified the timing of mass wasting events, determined long-term-averaged rates of cliff retreat and revealed the exposure history of shore platforms. In this contribution, we developed and explored a numerical model for the accumulation of [10]Be on eroding shore platforms. In a series of numerical experiments, we investigated the influence of topographic and water shielding, dynamic platform erosion processes, the presence and variation in beach cover, and heterogeneous distribution
of erosion on the distribution of [10]Be across shore platforms. Results demonstrate that, taking into account relative sea level change and tides, the concentration of [10]Be is sensitive to rates of cliff retreat. Factors such as topographic shielding and beach cover, act to reduce [10]Be concentrations on the platform, and may result in overestimation of cliff retreat rates if not accounted for. The shape of the distribution of [10]Be across a shore platform can potentially reveal whether cliff retreat rates are declining or accelerating through time. Measurement of [10]Be in shore platforms has great potential to allow us to quantify long-term
rates of cliff retreat and platform erosion.

## 1   Introduction

There is societal need to assess the rates, and the change of rates, at which cliffed coastlines will erode in the face of changing sea-levels and wave-climates that may result in more energetic coasts (Bray and Hooke, 1997; Trenhaile, 2010; Ashton et al., 2011; Barkwith et al., 2014). The lack of long-term records of cliff and shore-platform erosion rates is a key problem to
address (Trenhaile, 2014). An emerging tool to assess past rates of cliff retreat comes from the accumulation of cosmogenic radionuclides (CRNs), such as [10]Be, across active marine platforms that are generated via cliff retreat (Regard et al., 2012; Choi et al., 2012). CRNs have the potential to yield rates of cliff recession and shore-platform erosion over millenial times-scales. These records will provide long-term context for historical and present rates of cliff recession, and facilitate calibration of dynamic models of shore platform erosion and cliff recession (Trenhaile, 2000; Walkden and Hall, 2005; Ashton et al., 2011;



Matsumoto et al., 2016). The erosion of rocky coastlines is the result of a dynamic suite of processes that operate at the coast (Sunamura, 1992; Kennedy et al., 2014). The production of CRNs in the shore platform is in turn influenced by these processes and the resultant morphological evolution of the coast (Regard et al., 2012). This paper explores the factors that influence the spatial patterns of [10]Be across emergent platforms, with the objective of allowing us to better constraining quantitative

measures of cliff retreat rate.

Estimates of sea cliff retreat rates over decadal to centennial timescales have been made by observing the change in position of the cliff top, derived from historical maps, aerial photography, satellite imagery, and field surveys (e.g. Bray and Hooke, 1997; Costa et al., 2004; Dornbusch et al., 2006; Brooks and Spencer, 2010; Katz and Mushkin, 2013). Observations are limited by the length of historical records, which span at best the late 1800s to present. Sunamura (2015) demonstrated that longer-term

records are needed. The return period of coastal mass wasting events, the principal mechanisms of coastal cliff retreat, may in some cases be much longer than these historical records (Recorbet et al., 2010). Methods to estimate long-term rates of cliff retreat are required in order to time average rates across multiple failure events. It is important that modern observations of sea cliff retreat can be placed in the context of long-term coastal evolution, over time-spans unaffected by human intervention at the coast. Long-term records of cliff retreat in response to relative sea level (RSL) change are required in order to predict how

coastal erosion may proceed into the future in the face of anticipated RSL rise and increased storminess. Yet, by their very nature, eroding coastlines leave scant evidence of any former state, and their form reflects little about their long-term erosional trajectories (Matsumoto et al., 2016). Cosmogenic radionuclides (CRNs) have the potential to reveal the long-term history of coastal change and to quantify process rates along rocky coastlines (Recorbet et al., 2010; Choi et al., 2012; Regard et al., 2012; Rogers et al., 2012).

CRNs are isotopes produced by the interaction of cosmic rays with target elements in the upper few metres of the Earth surface. Measurements of the abundance of CRNs (particularly [10]Be) produced in rock and soil samples (*in situ*) provides a versatile geochronometer to quantify how long a sample has been exposed at/near the Earth surface, or interpret how rapidly it has been eroded (Balco et al., 2008; Dunai, 1995; Granger et al., 2013). Recently, these techniques have been used for application in rocky coast settings. Recorbet et al. (2010) used CRNs to demonstrate that a calcareous sandstone sea cliff in

south eastern France last failed around 3.5 ka (thousand years before present), suggesting a long return period for sea cliff retreat events at that site. Rogers et al. (2012) assessed long-term shoreline recession in Washington, USA by using CRNs to date large boulders, released and abandoned on the shore platform during recession of till bluffs.

The measurement of [10]Be concentrations from shore platform samples to estimate long-term rates of sea cliff retreat was pioneered by Regard et al. (2012) working on the flint-bearing chalk coastline near Mesnil-Val, France. Regard et al. developed

a numerical model to predict the concentration of [10]Be on shore platforms as a function of the rate of cliff retreat. They minimised residuals between model results and a transect of eight [10]Be concentration measurements from across the shore platform in order to estimate long-term average sea cliff retreat rates that were similar to estimates from historical observations; however, the uncertainties on the analyses were large and limited the resolution/confidence of the comparison.

The theoretical distribution of [10]Be concentrations on an eroding shore platform generally increases and then decreases with

35 distance from the modern cliff (Figure 1). In the nearshore, [10]Be concentrations increase offshore because the shore platform



has been exposed for longer. However, the rate of [10]Be production decreases offshore because cover by sea water attenuates the cosmic ray flux, hence the amount of cosmic radiation received by the platform decreases with increased water depth. Additionally, downwear of the coastal platform surface removes the highest concentration [10]Be rock from the surface and exhumes lower concentrations that were previously below the surface (Figure 1). The combined result of these factors is a

"humped" distribution of [10]Be concentrations. The magnitude of the maximum concentration is predicted to be proportional to the long-term averaged rate of sea cliff retreat (Regard et al., 2012); higher rates of sea cliff retreat result in lower [10]Be concentrations in the platform (Figure 1). Tides modify these predictions by altering the distribution of [10]Be production across the shore platform due to attenuation of cosmic rays in the water column. Regard et al. (2012) showed that increasing tidal range causes the cross-shore position of the peak concentration to migrate seaward. Regard et al. also demonstrated that whilst

rising RSL increases water depth, counter-intuitively it can result in higher [10]Be concentrations on the shore platform due to associated reduction in platform downwear rates.

Several other factors may influence the accumulation of [10]Be in shore platforms that have not been accounted for in previous coastal studies. Shore platforms may erode through gradual (e.g. abrasion) or episodic (e.g. quarrying) processes (Dornbusch and Robinson, 2011; Naylor et al., 2016), and the relative influence of these processes on the distribution of [10]Be has yet to

be explored. Similarly, the assumption that shore platforms evolve in morphological steady state (i.e. landward translation of a constant shore profile morphology through time that tracks relative sea level change), which is sometimes also referred to as equilibrium retreat, may not always be appropriate (Dickson et al., 2013). The style of coastal evolution is expected to influence the distribution of [10]Be across a shore platform. Modelling studies suggest that shore platform gradients may decline through time and platforms widen (Trenhaile, 2000; Walkden and Hall, 2005). Talus cones and beaches are often found fringing coastal

cliffs, covering shore platforms and therefore shielding them from cosmic rays. [10]Be concentrations in the shore platform would be reduced (Regard et al., 2012). Coastal cliffs may shield the platform from a portion of the incoming cosmic ray flux, thus also reducing [10]Be production in the nearshore, and shielding is proportional to cliff height.

In this study we quantified the sensitivity of platform CRN concentrations to topographic shielding, various processes of platform erosion/downwear, the presence/absence of beach cover, and transience in shore profile evolution. We addressed this

with a numerical model coupling cross-shore coastal evolution and [10]Be production to explore the potential for quantifying coastal retreat rates from [10]Be concentration measurements.

## 2   Numerical model for shore platform evolution

Cliffed, rocky coasts, are commonly fronted by shore platforms that are classified into two types (Sunamura, 1992). Type-A platforms are characterised by a gently sloping erosional platform surface extending offshore beyond maximum low water.

Type-B platforms are shallow gradient to sub-horizontal and terminate at their seaward edge at maximum low water through a scarp (Figure 2a). Numerical models of shore platform evolution have succesfully recreated both of these end-member morphologies (e.g. Trenhaile, 2000; Walkden and Hall, 2005; Matsumoto et al., 2016).



Numerical models of platform evolution demonstrate that shore platforms and adjacent sea cliffs tend towards a morphological steady state, such that coastal morphology does not change its form through time, and a constant cliff-platform geometry is translated landward through time. Under such conditions the morphology may reflect the combination of RSL change, tides and wave energy availability. However, the assumption of steady state retreat may not always be applicable (Dickson et al., 2013).

We expected the style of platform evolution to be important for the distribution of [10]Be across a shore platform. Therefore we performed experiments not only assuming steady state coastal retreat, but also using a dynamic model for platform evolution.

## 2.1 Steady State Coastal Retreat

We initially assumed that coastal cliff and shore platform evolution can be considered a steady-state process: a constant coastal cross-section profile is translated landward through time, with the elevation of the cliff-platform junction tracking RSL (Figure

2b). As such, platform downwear was assumed proportional to the product of cliff retreat rate and platform gradient $\alpha$ (Regard et al., 2012).

A steady state approach assumes platform downwear is gradual and constant (which implies abrasion is the dominant process), and proportional to the rate of cliff retreat. However, several coastal platforms have been observed to erode due to quarrying and block removal (e.g. Dornbusch and Robinson, 2011; Naylor et al., 2016). In order to explore the potential impli-

15 cation of these erosion processes for the accumulation of [10]Be in a shore platform, we also evolved a series of stepped platforms by steady state retreat.

## 2.2 Numerical Model for Dynamic Platform Evolution

In order to explore [10]Be concentrations across a transient (i.e. not steady state) shore platform, we developed a simple numerical model for shore profile evolution (the ROck and BOttom COastal Profile [RoBoCoP] Model), broadly similar to those of

20 Sunamura (1992), Anderson et al. (1999), Trenhaile (2000) and Walkden and Hall (2005). These models assume that horizontal erosion at the water level is proportional to the availability of wave energy (or by proxy, wave height). The shore profile was considered as a regularly spaced vertical stack of cells with horizontal position $x$ [L] (all dimensions denoted in square brackets as [$L$]ength, [$M$]ass and [$T$]ime). The change in position of the coast at the water level $x_w$ [L] through time $t$ [T] was assumed to be linearly proportional to the height of breaking waves reaching the shore $H_c$ [L]:

$$\frac{dx_w}{dt} = K \rho_w \, g H_c \tag{1}$$

In Equation 1, $K$ [L$^2$·T·M$^{-1}$] is a coefficient related to the resistance of bedrock to erosion, $\rho_w$ [M·L$^{-3}$] is the density of water and and $g$ [L·T$^{-2}$] is acceleration due to gravity. The water depth $h_w$ [L] for the initiation of wave breaking $h_b$ [L] was related to wave height $H$ [L], such that breaking wave height $H_b$ [L] is the wave height that exceeds a water depth-dependent threshold:

$$H_b = 0.78 \, h_b \tag{2}$$





Breaking wave height was determined by iterating deep water wave conditions from deep to shallow water and calculating $h_w$ according to linear wave theory (e.g. Hurst et al., 2015), or following an empirical relationship relating breaking wave conditions to offshore wave conditions (e.g. Komar and Gaughan, 1972):

$$H_b = 0.39 \, g^{1/5} \, T^{2/5} \, H_0^{4/5} \tag{3}$$

In Equation 3, $T$ [T] is wave period and $H_0$ [L] is deep water wave height. If Equation 3 is used to predict $H_b$, Equation 2 can be inverted to determine the water depth at which wave breaking begins. Following wave breaking, wave height is assumed to decay exponentially with distance across the platform, such that the wave height at the water line can be described as:

$$H_c = H_b \, \mathrm{e}^{-k \, W_s} \tag{4}$$

In Equation 4, $W_s$ [L] is the width of the surf zone, measured from $x(h_w = h_b)$ to $x(h_w = 0)$, $k$ is a dimensionless constant
that represents the rate of breaking wave energy dissipation, which reflects bed roughness in the surf zone (Trenhaile, 2000); we used $k = 0.02$ throughout. These properties are all time dependent, because the elevation of the water surface varies due to tides (superimposed onto any RSL change). The distribution of erosion across the platform was integrated across the tidal cycle of period $T_t$ [T]. Platform erosion below the water line was assumed to decline exponentially with water depth. Combining Equations 1-4, the governing equation for evolution of the shore platform becomes:

$$15 \quad \frac{dx}{dt} = K \rho_w \, g \int_{t=0}^{T_t} H_b(t) \, \mathrm{e}^{-(k \, W_s(t) + h_w(t))} \qquad \{h_w \in \mathbb{R} : h_w \geq 0\} \tag{5}$$

### 2.3   Beach Cover

In order to explore the influence of beach cover on the accumulation of [10]Be on the coastal platform, we approximated the profile morphology of beaches using a power-law function (Figure 3; Bruun, 1954):

$$z_b = z_{0b} - A x_b{}^m \tag{6}$$

In Equation 6, $z_b$ [L] is the elevation of the beach, $z_{b0}$ [L] is the elevation of the beach at the top of the berm, $A$ [L$^{1/3}$] is a scaling parameter that relates to the size of beach material, and $m$ is a dimensionless exponent that represents the distribution of wave energy dissipation on the shoreface. We used a value of $A = 0.12$ suitable for gravel, and a shape exponenet $m = 2/3$, consistent with a number of studies (Dean and Darlymple, 2002). The beach profile extends from the position of top of the berm described by the beach width $B_w$ [L] and berm height $B_h$ [L] (Figure 3).



## 3 Numerical model for $^{10}$Be Production in the Shore Platform

The production of $^{10}$Be fundamentally depends on how long the surface has been exposed, and the rate at which material is removed through erosion (Gosse and Phillips, 2001). On a shore platform, exposure is modulated by topographic shielding, beach cover and water cover (Regard et al., 2012). Erosion of the shore platform may take place through abrasion or plucking, and may not be spatially and temporally uniform (Dickson et al., 2013).

### 3.1 Production of $^{10}$Be in Rock

The concentration of $^{10}$Be $N$ [atoms M$^{-1}$] changes through time according to:

$$\frac{dN}{dt} = P(h_r) - \lambda N \tag{7}$$

In Equation 7, $P(h_r)$ [atoms·M$^{-1}$·T$^{-1}$] is the depth-dependent production rate of $^{10}$Be , $h_r$ [L] is depth below the rock surface, and $\lambda = 4.99 \times 10^{-7}$ is the $^{10}$Be radioactive decay constant (Chmeleff et al., 2010; Korschinek et al., 2010). The production of $^{10}$Be *in-situ* at and near the Earth surface declines exponentially with depth (self-shielding) as the cosmic ray flux attenuates (e.g. Gosse and Phillips, 2001; Balco et al., 2008; Mudd et al., 2016):

$$P(h_r) = \sum_i P_s(i) \mathrm{e}^{-\left(\frac{h_r}{h^*(i)}\right)} \tag{8}$$

In Equation 8, $P_s$ [atoms·M$^{-1}$·T$^{-1}$] refers to the production rate at the rock surface for the production pathway $i$. $^{10}$Be is predominantly produced by neutron spallation, with minor contribution from fast and slow muon interactions. Muogenic production penetrates much deeper into the Earth surface and therefore may source a significant part of observed $^{10}$Be concentrations. Similar to West et al. (2014), we used a single exponential curve to integrate fast and slow muogenic production pathways (Braucher et al., 2011, 2013). The attenuation of spallation and muogenic reactions declines according to an attenuation length scale $h^*$ [L] that is dependent on the density of bedrock $\rho_r$ [M·L$^{-3}$] and a pathway dependent attenuation factor $\Lambda$ [M·L$^{-2}$] (for spallation $\Lambda = 1600$ kg m$^{-2}$; and for muogenic production $\Lambda = 42000$ kg m$^{-2}$):

$$h^*(i) = \frac{\rho_r}{\Lambda(i)} \tag{9}$$

### 3.2 Topographic Shielding

Rock surfaces may be shielded from a portion of the incoming cosmic ray flux by local topography. At the coast, sea cliffs can block a significant portion of the sky and therefore partially shields the platform from cosmic rays. Typically shielding factors can be quantified by field surveys (Dunne et al., 1999) or from topographic data (Codilean, 2006; Mudd et al., 2016). However, when the sea cliff is retreating, the amount of topographic shielding at a fixed location will change through time with increasing distance to the position of the cliff.





In order to model the influence of topographic shielding by the cliff on the accumulation of $^{10}$Be, we idealised that the cliff is vertical and of height $C_H$ [L], and assumed the cliff line to be straight in planform. For a given distance from the cliff $x_c$ we defined the viewshed generated by the cliff as the angle in the sky $\theta$ from horizontal made by the top of the cliff in the azimuth direction $\phi$ for each observation $j$:

$$\theta(j) = \tan^{-1}\left(\frac{C_H \cos\phi(j)}{x_c}\right) \tag{10}$$

Following Dunne et al. (1999), a shielding factor $S_{cliff}$ is the ratio of cosmic ray flux given the viewshed $F$ to the maximum cosmic ray flux for an unobstructed flat surface $F_{max}$:

$$S_{cliff} = \frac{F}{F_{max}} = \frac{\Delta\phi}{2\pi} \sum_{j=1}^{n} \sin^{m+1}\theta(j) \tag{11}$$

where $n$ is the total number of viewshed observations, that span the azimuth range $\phi = 0 - 360^o$. A straight, vertical cliff line yields $S_{cliff} = 0.5$ at $x_c = 0$ since 50% of the sky is obstructed.

Topographic shileding can also be modelled explicity from a digital elevation model (DEM) following Codilean (2006). However these shielding values only apply to the current platform and cliff morphology. We compared Equation 11 to shielding factors calculated from a DEM for shore platforms on the coast of East Sussex, UK (1 m resolution airborne LiDAR; data courtesy of Channel Coast Observatory; www.channelcoast.org; accessed 25th May 2014), using software published by Mudd et al. (2016) [https://github.com/LSDtopotools/LSDTopoTools_CRNBasinwide].

### 3.3 Water Shielding

A shore platform is periodically exposed or submerged by the sea due to tides. The cosmic ray flux attenuates exponentially with water depth $h_w$ [L] so that the production rate at the surface of the platform decreases according to:

$$P_s(x,i) = P_0(i)e^{\frac{-h_w(x)}{h_w^*(i)}} \tag{12}$$

In Equation 12, $P_0$ [atoms·M$^{-1}$·T$^{-1}$] is the reference production rate at the water surface, which is a function of latitude and altitude (e.g. Lal, 1991; Dunai, 2000; Stone, 2000; Desilets et al., 2006; Lifton et al., 2005). Equation 9 is used to calculate $z_w^*$ by substituting water density $\rho_w$ [M·L$^{-3}$] for $\rho_r$. Thus, the production by each pathway can be adjusted for attenuation in the water column, however this adjustment depends on the tidal cycle at the site of interest.

### 3.3.1 Tides

Tides modify production in the platform by varying water depth $h_w$ and intermittently exposing the platform sub-aerially. Regard et al. (2012) demonstrated that tides have a net effect to reduce $^{10}$Be production in the upper inter-tidal platform due to periodic platform submergence that reduces the net cosmic ray flux received, while $^{10}$Be production in the lower platform





increases relatively due to periodic exposure. Here we extended this analysis to explore the influence of tidal regime on $^{10}$Be production.

Predictions of the tide are made as the sum of its harmonic constituents (e.g. Pugh and Woodworth, 2014):

$$z_w(t) = \sum_n H_n \cos(\sigma_n t - g_n) \tag{13}$$

In Equation 13, $z_w$ [L] is the elevation of the mean water surface, the subscript $n$ refers to the tidal constituent, $H_n$ is the amplitude [L] of that constituent, $\sigma_n$ is the angular speed [º·T$^{-1}$], and $g_n$ is the phase lag [T]. Predictions of tidal elevation can be converted to water depth across the platform:

$$h_w(x,t) = \begin{cases} z_w(t) - z_r(x,t) & \text{if } h_w \geq 0 \\ 0 & \text{if } h_w(t) < 0 \end{cases} \tag{14}$$

In Equation 14, $z_r$ [L] is the elevation of the platform surface; $h_w$ must be positive (i.e. if the water level falls below the

platform elevation the water depth at this location is zero). Combining Equation 12 and 14, the production rate at the platform surface averaged over a tidal cycle (T) can then be calculated as:

$$P_s(x,i) = \frac{1}{T} \sum_{t=0}^{T} P_0(i) e^{\frac{-h_w(x,t)}{h_w^*(i)}} \tag{15}$$

We used Equations 13-15 to compare the distribution of platform surface production rates across the shore platform for hypothetical diurnal, mixed and semi-diurnal tidal regimes, generated using the tidal constituents listed in Table 1. These

hypothetical tidal regimes were designed to cover a similar range in water levels across roughly a single lunar duration. We also compared the predicted $P_s$ for a simple semi-diurnal tide to the effective production averaged over a full year of tidal records from the a tide gauge at Newhaven, East Sussex [available from www.channelcoast.org; accessed 2$^{nd}$ October 2014].

### 3.3.2   Relative Sea Level Change

For model experiments driven by steady state retreat, RSL rise results in higher concentrations of $^{10}$Be across the shore plat-

form. This may seem counter-intuitive at first, however less vertical downwear of the platform surface is required to maintain the steady state profile during rising RSL and less material with high $^{10}$Be concentrations is removed by downwear (Regard et al., 2012). In this study, we used constant rates of RSL change, since Holocene sea-levels have been relatively stable over the last 7 ka; however, for application to specific sites, RSL histories derived from local sedimentary records or regional crustal flexure models (e.g. Bradley et al., 2011) are recommended.



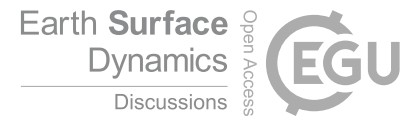

## 4   Experimental Setup

We conducted a number of experiments to explore the influence of specific processes on the distribution of [10]Be across a shore platform. Here we describe the model setup used to investigate (i) the influence of platform erosion process; (ii) the influence of beach cover; (iii) the influence of tides; and (iv) the influence of transient profile evolution on the accumulation of [10]Be in the shore platform. Global parameters consistent across all experiments are listed in Table 2.

### 4.1   Block Removal Processes

To investigate the influence of block removal processes, we evolved a steady state profile with a fixed average slope $\alpha$ = 1/100 consisting of offset horizontal surfaces akin to horizontal bedding planes. The separation between surfaces was varied from 0 m (uniform sloped platform, downwear inferred to be by abrasion) to 0.8 m. Steps in the steady state profile were allowed to migrate landward through time in concert with a prescribed constant cliff retreat rate of $\epsilon = 0.1$ m yr$^{-1}$. RSL was held constant and a diurnal tide with $H_n$ = 1.0 m was used.

### 4.2   Beach Cover

Beaches cover the shore platforms in the nearshore, which partially shields the platform from cosmic rays and, as a result, reduces [10]Be production in the platform surface. The presence of beach cover may not be consistent, but will depend on the supply of material from the adjacent cliffs, the supply and removal of sediment due to both alongshore and cross-shore sediment transport, and the rate at which beach material is physically and chemically weathered. We conducted a series of experiments designed to investigate the influence of beach cover on the accumulation of [10]Be on a shore platform. Firstly, we considered a simple case where beach morphology is held constant through time, and explore the influence othat constant beach width ($B_w$ = 0, 10, 20, 50, 100 m) on the concentration of [10]Be in the shore platform. Secondly, we considered the condition that beach width may vary through time, with beach width varying as a sinusoidal function over decadal timescales (wavelength = 100 years), with an average $B_W$ = 50 m and amplitude of 30 m. Thirdly, we considered the condition that beaches may be being lost through time, as beach material is transported away more rapidly than it is supplied by alongshore transport and cliff erosion. These scenarios all evolved the shore profile by steady state retreat with a cliff retreat rate of $\epsilon = 0.1$ m yr$^{-1}$. RSL was held constant and a diurnal tide with $H_n$ = 1.0 m was used.

### 4.3   Steady state and Transient Shore Platform Evolution

Previous CRN studies have assumed that shore platform evolution proceeds in steady state such that a constant shore profile morphology is translated landward through time, and tracks the trajectory of RSL change (Regard et al., 2012). Here, we explored the influence of RSL rise on shore platform morphology and the resulting distribution of [10]Be across shore platforms with RoBoCoP (see Section 2.2). We compared the results of dynamic shore platform evolution experiments to those of steady state shore profile retreat to investigate the influence that the assumption steady retreat may have on the accumulation of [10]Be and our ability to interpret rates of cliff retreat. The initial and boundary conditions for the model were held constant (see Table





3). The model is initialised with a sloped platform profile and forward modelled for 10 Ka. In our first set of experiments, we held RSL constant whilst during the second set of experiments RSL rose at a constant rate (0.5 mm yr$^{-1}$). RSL rise is known from previous experiments to influence the rate of cliff retreat and how it changes through time, the morphology of the platform, and the amount and distribution of platform downwear (Trenhaile, 2000; Walkden and Hall, 2005). We compared

the resultant $^{10}$Be concentrations to predictions that assume steady state retreat in order to determine whether this assumption allows estimatation of average transient retreat rates.

## 5    Results

### 5.1    Topographic Shielding

The topographic shielding scaling factor $S_T$ is predicted to have a value of 0.5 immediately adjacent to the cliff, increasing

nonlinearly with distance from the cliff. (Figure 4a). Cliff height principally controls the rate at which $S_T$ increase, such that taller cliffs result in a greater degree of shielding for a greater distance offshore. $S_T$ approaches unity asymptotically by $x_c \approx 4 \cdot C_H$; there is little influence of the cliff on $^{10}$Be production beyond. Our approach (Equation 11) assumes a straight cliff line with constant cliff height. We compared this approach to the distribution of topographic shielding factors modelled for a real stretch of coastline in East Sussex, UK, following Codilean (2006) (Figure 4b). Again, these ditributed $S_T$ values

are smallest immediately adjacent to cliff ( 0.5) and increase nonlinearly with distance across the shore platform. The values calculated from the DEM are in good agreement with Equation 11 as demonstrated by three example transects, each with different cliff heights (Figure 4c). Differences between observed and model $S_T$ values are due to variation in the gradient of the cliff face and the planform geometry of the cliff line. Since it is unlikely in most settings to have information on how $C_H$ and the planform geometry may have changed through time, we used Equation 11 to represent topographic shielding, assuming

constant $C_H$.

### 5.2    Water Shielding by Tides

Variation in water level at hourly timescales due to tides modifies the cosmic ray flux delivered to the platform. The tidal regime at a site may influence distribution of $^{10}$Be production across a shore platform, and so we explored the effects of hypothetical diurnal, mixed and semi-diurnal tidal regimes (Figure 5a-f) on $^{10}$Be concentrations. The tidal constituents used to model these

are shown in Table 1. Figure 5g shows the distribution of relative spallation production $P_s/P_0$ across a planar platform defined by $\alpha = 1/100$ under these six different hypothetical tidal regimes. Tides act to reduce the production in the nearshore/upper intertidal zone due to periodic submergence of the upper platform, whilst production in the lower intertidal zone is increased due to periodic exposure of the platform. Simple diurnal/semidiurnal tides show the strongest modification of $^{10}$Be production according to our experiments, but this is strongly dependent on $H_n$ for the additional constituents. Different tidal regimes may

result in differences in the production of $^{10}$Be across the coastal platform and therefore it is important to consider the tidal regime at a particular site to model $^{10}$Be production in the platform.



Figure 6 shows the distribution of the effective production rate at the platform surface for purely diurnal tides with an amplitude of 2.4 m, compared to the tide gauge water level record at Newhaven, East Sussex, UK. Incorporation of a tidal record with the full range of harmonic components dampens production very slightly in the nearshore region and slightly increases production further offshore relative to a simple the semi-diurnal tide model. These differences do not significantly

influence [10]Be production. A simple diurnal/semidiurnal tide model with a single representative tidal amplitude is appropriate at some sites.

## 5.3   Block Removal Processes

Model results of the migration of bedrock steps due to block removal processes across a shore platform that evolves through steady state retreat is shown in Figure 7a for a range of step sizes. The landward migration of steps results in the sudden

exposure of platform that was previously the depth of the step size below the platform surface. Since [10]Be production rates decline exponentially with depth into the rock (Equation 8), block removal unearths rock with significantly lower concentration, resulting in sudden drops in [10]Be concentration with distance across the platform (Figure 7b). However, the flat-topped steps are subject to no vertical downwear in these simulations and so they continue to accumulate [10]Be such that, between bedrock steps, concentrations increase with distance from the cliff. The result is a saw-toothed distribution of concentrations, superimposed

on the "humped" distribution expected when there is no block removal. The magnitude and wavelength of the variability in concentrations is controlled by the size and frequency of the bedrock steps.

## 5.4   Beach Cover Control on [10]Be Concentrations

Beaches were represented by a Bruun Profile (Equation 6) that extends seaward from a prescribed beach width and berm height (Figure 3). Beach cover may not be constant through time. We explored the influence of beach width, variable beach width,

and declining beach width on shore platform [10]Be concentrations.

### 5.4.1   Influence of Beach Cover

Beach cover is expected to partially shield the nearshore platform from cosmic rays and therefore reduce concentrations of [10]Be in the platform surface. Wider beaches should result in lower [10]Be concentrations. We ran model experiments with different values of $B_W$, which was held constant for the duration of each experiment ( Figure 8a). The predicted distributions of [10]Be

concentrations increase and then decrease in the offshore direction, and the magnitude of the "hump" is reduced when beaches are wider (Figure 8b). The position of the peak with respect to the cliff moves offshore as production in the nearshore is reduced by beach cover. The black line corresponds to the scenario with no beach cover. The concentrations of [10]Be at the platform surface decrease in a linear fashion with wider beaches (Figure 9), to the extent that a 50 m wide beach results in a 18% reduction in the magnitude of the [10]Be concentrations. If not accounted for, this would lead to over prediction of cliff retreat

rates when inverting measured [10]Be concentrations, since, in the absence of beaches, lower concentrations of [10]Be suggest faster cliff retreat rates (Regard et al., 2012).





### 5.4.2   Influence of Variable Beach Cover

Beach cover may not be constant through time, however there is rarely any evidence available revealing how beaches may have changed during the Holocene on eroding coastlines, since eroded material tends to be removed. We explored how variable beach cover modifies [10]Be concentrations by modelling $B_W$ through time as a sinusoidal function with wavelength 100 years and amplitude 30 m about an average of $B_W$ = 50 m (Figure 10a). Thus, the beach progrades and regresses every 100 years, maintaining an equilibrium beach profile (Figure 10b). The resultant distribution of [10]Be concentrations are compared to simulations with no beach cover and a constant $B_W$ = 50 m (Figure 10c). We found that there is very little difference in [10]Be concentrations between a constant and variable beach width scenario.

We also explored scenarios in which beach cover was reduced through time. Figure 11 shows the morphological evolution and concentrations of [10]Be in the shore platform surface predicted for the condition where beaches have thinned from a constant width of 50, 100 or 200 m over the last 1000 years. The thinning of beaches, and thus the presence of more beach cover in the past reduces the expected concentrations of [10]Be in the shore platform. This will result in over-prediction of cliff retreat rates when not accounted for, since platform [10]Be concentrations will be lower, as otherwise associated with more rapid rates of cliff retreat.

### 5.5   Transient Shore Platform Evolution

### 5.5.1   Constant sea level

We modelled the evolution of a shore profile given constant RSL and the consequential concentrations of [10]Be across the modelled shore profile. The gradient of the shore platform decreases through time, as the platform widens (Figure 12a). A stepped platform emerges with the step at the lower tidal limit, reflected by a downward step in [10]Be concentrations on the shore platform (Figure 12b). Wave energy attenuation increases as the platform widens such that cliff retreat rates decline as the model simulation proceeds (Figure 12c). Reduction in the mean gradient of the platform is shown in Figure 12d. The distribution of [10]Be concentrations is humped, similar in form to predictions of a steady state retreat model (Regard et al., 2012). The magnitude of the hump increases through time, as cliff retreat rates slow.

In order to compare results directly, we then ran simulations where the morphology evolved in steady state (cliff retreat rate was held constant, using the median values reported in Figure 12b, platform gradient was taken as the instantaneous average platform gradient in each profile in Figure 12a (measured between platform elevations $z$ = 0.5 and $z$ = -1 m). These simulations ran for long enough that 1000 m of cliff retreat had occurred. Comparison of the distribution of [10]Be concentrations between the transient (solid lines) and steady state (dashed lines) model runs is shown in Figure 13a, shaded by their retreat rate as listed in the legend in Figure 12b. The magnitude of the peak in concentrations were in this case similar between transient and steady state simulations (Figure 13b), however, the position of the peak was further offshore in the steady state simulations (Figure 13c).





### 5.5.2 Rising Relative Sea Level

We modelled the transient evolution of a shore profile and [10]Be concentrations with sea level rise of 0.5 mm yr[-1] (Figure 14). The shore platform evolves rapidly to form a low gradient shore platform ramp (Figure 14a). The elevation of the platform/cliff junction increases through time and tracks the high tide level, superimposed on the trajectory of RSL rise. Cliff retreat rates

are initially rapid and decline towards a constant rate of 0.32 cm yr[-1] (Figure 14c). Having attained a constant rate of cliff retreat, concentrations of [10]Be in the platform as a function of distance from the cliff are constant, consistent with the concept of steady-state morphological retreat. Due to low platform gradients, and the absence of significant platform downwear below the intertidal zone, concentrations of [10]Be continue to increase offshore and so no "hump" in concentrations is observed.

We compared these predictions to those of a steady state retreat model with cliff retreat rate (32 cm yr[-1]) and platform gradient

(1/500) observed in the above transient experiment. The results are plotted as the dashed line in Figure 14b. The steady state retreat model similarly predicts concentrations that increase across the entire shore platform due to minimal platform downwear in the presence of sea level rise. However, the steady state model predicts significantly higher concentrations of [10]Be across the shore platform, by about a factor of two, due to differences in the distribution of downwear between the two models. In the transient model runs, the amount of platform downwear declines exponentially with water depth, leading to rapid erosion in

the mid to upper intertidal zone, where [10]Be production rates are most rapid. This is evidenced by the concave-up shore profile nearest to the cliff. Therefore not only does more downwear occur on the upper part of the transient profile, but as a result, water depths increase more quickly offshore, and increased water shielding reduces [10]Be production and concentrations on the shore platform as a whole. Thus, if we were to assume steady state retreat and a planar shore platform in a setting where there is more platform downwear in the intertidal zone, we are likely to overestimate retreat rates since [10]Be concentrations will be

lower than expected for a given retreat rate.

## 6 Discussion

We have identified that concentrations of [10]Be on shore platforms are sensitive to (i) shielding by cliffs; (ii) the type of process (abrasion vs quarrying); (iii) the tidal regime; (iv) the nature of beach cover and how it has changed through time; and (v) the style (steady state *vs.* transient) by which the platform evolves, particularly in the face of RSL change.

Nevertheless, these results demonstrate that there is great potential for CRN measurements to provide first-order estimates of long-term rates of sea cliff retreat (Regard et al., 2012; Rogers et al., 2012; Choi et al., 2012). Quantifying these factors, and how they may have changed over the millennial timescales required by CRN studies may not always be possible. Some of these factors can be accounted for explicitly in site specific studies, such as cliff shielding, given the height of the modern cliff, tides, informed by nearby tide gauges, and RSL change, from proxy records or glacio-isostatic adjustment models. We are still

required to assume that these factors have not changed through time; it is always necessary, to some extent, to extrapolate the modern coastal configuration back into the past. We have little or no information about the paleaotopography offshore from a modern cliff. Observations of beach cover can be made from historical data (e.g. Dornbusch et al., 2008), but are limited to the length of the historical record (c. 150 years maximum) and may be influenced by human intervention at the coast. Many of these





factors act to reduce predictions of [10]Be in the shore platform, but maintain the overall "humped" shape of the distribution. It appears not possible to distinguish from the [10]Be concentrations what the history of beach cover is if different from present. It may be possible to identify when a platform is evolving transiently (see below). If factors that reduce [10]Be concentrations were not considered then we would expect to overpredict rates of cliff retreat. Consequently, cliff retreat rates derived from CRN

studies might be considered maximum estimates. Below we discuss some of these issues further in the context of quantifying long-term rates of cliff retreat.

## 6.1   Block removal processes

Our simplified experiments demonstrate that where block removal processes are an important process for platform evolution, the expected distribution of [10]Be becomes more variable, with higher frequency variation superimposed on the expected

'humped' distribution (Figure 7). Consequently, CRNs can still reveal rates of cliff retreat, but a careful sampling strategy will be required to account for block removal processes. Observations of platform erosion process should be made when sampling for CRNs, and the size of steps or blocks will be important to record. Where available, information on the rate at which steps migrate and blocks are removed will help to inform sampling and interpretation (e.g. Dornbusch and Robinson, 2011; Naylor et al., 2016). Measuring [10]Be concentrations on shore platforms where block removal processes are dominant may allow the

rate of step migration to be determined in the presence of large bedrock steps (Figure 7). This will require high density sampling, focused on an individual bedrock step surface. If sampling on a platform with a stepped profile, data on the size of steps should be recorded and position of samples relative to steps will also be important.

## 6.2   Transient shore platform development

Previous studies had assumed steady state profile retreat was an adequate description of the morphological evolution of the

shore platform in order to predict cliff retreat rates from [10]Be concentrations in the shore platform over millennial timescales (Regard et al., 2012). Dynamic shore profile evolution models (e.g. Sunamura, 1992; Anderson et al., 1999; Trenhaile, 2000; Walkden and Hall, 2005; Matsumoto et al., 2016) predict that coasts tend towards steady state whereby rapid cliff retreat widens shore platforms and the resultant increased wave energy dissipation reduces cliff retreat rates and increases erosion of the shore platform. We coupled predictions of [10]Be production to a dynamic shore profile evolution model (RoBoCoP; see

section 2.2) in scenarios with (i) constant RSL, such that cliff retreat rates gradually reduced through time due to widening of the shore platform; and (ii) rising RSL, such that cliff retreat rates tended to a constant rate in time.

In scenario (i) the concentrations of [10]Be on the shore platform increased through time as cliff retreat rates declined (Figure 12). Comparison to predictions of steady state profile retreat (Figure 13) revealed that the position of the peak in [10]Be concentrations was further offshore when assuming steady state retreat. The results of Regard et al. (2012) using a steady state retreat

model demonstrated that the position of the peak is sensitive to the tidal range and the platform gradient. Tidal range was held constant for all of our simulations, however platform gradient declined during transient platform evolution simulations. Comparison of field measurements of [10]Be concentrations and model runs that assume steady state retreat may therefore reveal when platform gradients are declining through time based on a mismatch in the position of the observed and modelled peak



concentrations. In this case, the magnitude of the peaks suggests similar cliff retreat rates whether assuming steady state or transient shore profile evolution.

In scenario (ii) cliff retreat rates tended toward a constant rate with the result that the distribution of [10]Be was approximately constant through time (Figure 14). The concentrations of [10]Be were not consistent with a steady state evolution scenario in which the platform gradient is fixed, which predicted roughly twice the amount of [10]Be for a particalar position on the platform. The difference can be explained by the dissimilar platform morphology brought about by uneven distribution of platform downwear in the transient model simulations. Greater rates of downwear in the intertidal zone lower the platform more rapidly in the nearshore, which removes [10]Be-laden rock and results in deeper water in the nearshore (and therefore reduced [10]Be production) than in the steady state model runs that assume constant $\alpha$. In attempting to reconstruct cliff retreat rates the elevation profile of the platform should be accounted for such that the distribution of downwear across the platform is taken into account. This could be aided by data on the distribution of downwear rates from micro-erosion-meter measurements (e.g. Robinson, 1977; Porter et al., 2010; Stephenson et al., 2010, 2012) Failure to do so would result in overprediction of erosion rates since faster retreat rates would be required to match the lower concentrations observed on the platform.

## 6.3 Beach cover

Beach cover in the nearshore partially shields the underlying platform from [10]Be accumulation and results in a reduction in the magnitude of the hump. Yet we suggest that the significant reductions are only observed for relatively high beach widths ($\geqslant 50$ m), that are only likely to persist on slowly eroding coastlines, since they may absorb wave energy and partially protect the cliffline. Theoretical considerations suggest a dynamic relationship between beach material and cliff retreat; beaches can act both to provide abrasive tools to enhance cliff erosion or to protect the cliff from wave energy (e.g. Limber and Murray, 2011). We did not model this dynamicism, but instead favoured exploration of simple relationships between beach dynamics and the accumulation of CRNs in the platform in order to try and understand first order controls. We found that variation in beach cover through time was not important for CRN concentrations on a shore platform when compared to a scenario with a constant and representative average beach width. However, our approach treated cliff retreat rate as constant thus not capturing feedbacks between beach cover and cliff retreat.

## 6.4 Inheritance of [10]Be

Predicted concentrations of [10]Be in shore platforms are relatively low compared to typical applications in geomorphic studies. Cosmogenic [10]Be produced at the surface is dominated by spallation reactions, but muogenic production penetrates deeper into the Earth surface (Heisinger et al., 2002b, a; Braucher et al., 2013). Muons only account for a small fraction of the production once the platform is exposed, but platforms may already contain an appreciable concentration of [10]Be prior to exposure formed by deep-penetrating muons over much longer timescales. The amount of "inherited" [10]Be will decline with the depth below the top of the cliff (i.e. the cliff height) but is also influenced by the rate of surface lowering at the cliff top (Lal, 1991). It is important to collect nearshore samples to quantify how much inherited [10]Be is in the rock prior to platform exposure.



Platforms may also be geomorphically inherited landforms, having formed during a previous inter-glacial sea level high stand (e.g. the Eemian; 130-115 ka) and reoccupied by the sea during the Holocene (e.g. Trenhaile, 2001; Chao et al., 2003; Choi et al., 2012). If shore platforms are contemporary features, [10]Be concentrations will be low, and their distribution controlled primarily by cliff retreat rate, and other factors explored in this paper. If shore platforms are inherited features then [10]Be

concentrations will be substantially higher reflecting subaerial exposure during the last glacial period Choi2012. The location of a sudden increase in [10]Be concentrations on the shore platform may reveal the location of the paleao-cliffline formed the previous time the shore platform was occupied, allowing a long-term average cliff retreat rate to be determined from the difference to the modern cliff position (Regard et al., 2012).

### 6.5  Interpreting rates of cliff retreat from [10]Be concentrations

In order to determine rates of cliff retreat from [10]Be concentrations on shore platforms, results of a coupled morphological and CRN production model are compared to measured concentrations in order to statistically determine the most likely combinations of parameters and retreat rates that yield close fit between modelled and observed concentrations (Regard et al., 2012). In addition to high rates of cliff retreat, topographic shielding (Figure 4) due to adjacent sea cliffs and the presence of beaches (Figure 8) that may have previously been more extensive (Figure 11) all act to reduce the amount of [10]Be in the shore

platform, compared to a scenario with no cliffs and no beaches. Therefore, not accounting for the presence of beach material and topographic shielding is likely to lead to overestimation of rates of sea cliff retreat. Regard et al. (2012) didn't account for beach cover and how it may have changed through time, but observed beaches are relatively narrow and thin, and the cliffs at their field site were small enough to have negligible effect on their estimated cliff retreat rates, given uncertainties in their [10]Be concentration measurements.

Observations of beach widths and berm heights may be made from both modern and historical data (e.g. Dornbusch et al., 2008), but there is little information about how beaches may have changed over millennial timescales. However, exposed shore platforms are most likely to be associated with locations with little or declining beach cover, since if beaches were accumulating the platform would not be exposed and cliff retreat rates might be expected to drop. The model predicts minimal differences in the distribution of [10]Be concentrations in the shore platform surface for simulations with constant versus variable beach cover

(Figure 10). A representative average beach width guided by historical observations may suffice when interpreting cliff retreat rates from [10]Be concentrations.

### 7  Conclusions

We find that the accumulation of [10]Be in the shore platform is primarily sensitive to the rate of cliff retreat. Concentrations of [10]Be in the shore platform are also influenced by a number of other factors, including topographic shielding by sea cliffs,

shielding due to beach and talus cover, water shielding due to tides and relative sea level change, the type of processes eroding the platform, and the style of platform evolution (steady state vs transient). These factors generally tend to reduce the production of [10]Be in the shore platform, particularly in the nearshore, nearest to cliffs and where the platform is most likely to be covered





by a beach. Nevertheless, comparison of measured $^{10}$Be concentrations to model simulations that include these factors should allow determination of long-term average cliff retreats. If these factors are not adequately considered then there will be a tendency to overpredict cliff retreat rates and so cliff retreat rates derived from cosmogenic $^{10}$Be might be considered as maximum estimates. The shape of the distribution of $^{10}$Be across a shore platform can reveal whether cliff retreat rates are

5  declining or accelerating through time. We conclude that measurement of $^{10}$Be in shore platforms has great potential to allow us to quantify long-term rates of cliff retreat and platform erosion.

*Author contributions.*  MDH, DHR and MAE concieved the study; MDH wrote the software and performed the analyses; MDH wrote the manuscript with contributions from all authors.

*Acknowledgements.*  The topographic and tide data used in this paper is available from the Channel Coast Observatory at www.channelcoast.

10  org. All code used in this analysis is open source and can be downloaded from https://github.com/mdhurst1/RoBoCoP_CRN. We are grateful to Robert Anderson, Uwe Dornbusch, Zuzanna Swirad and Nick Rosser for discussion that shaped the development of this study.



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





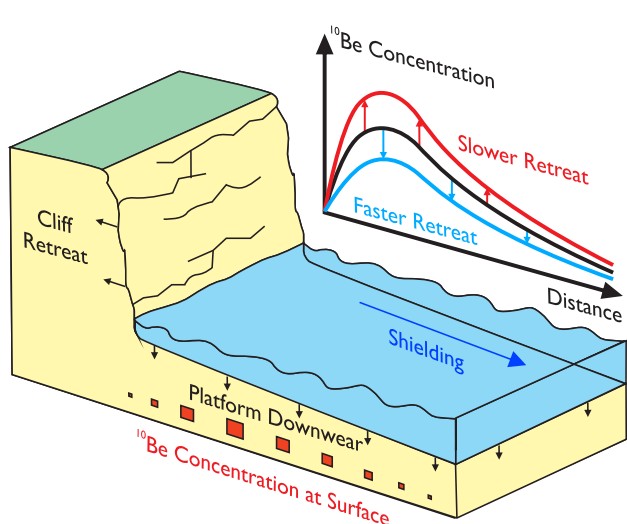

**Figure 1.** Schematic diagram of a rocky coast showing the expected distribution of $^{10}$Be across a shore platform. Cliff retreat exposes pristine platform with low $^{10}$Be concentrations. Exposure time increases with distance from the cliff (increasing $^{10}$Be concentrations), but platform downwear removes $^{10}$Be-rich rock, and increased water shielding reduces $^{10}$Be production offshore. The result is a hump-shaped distribution where the magnitude of the hump is inversely proportional to rate of cliff retreat.





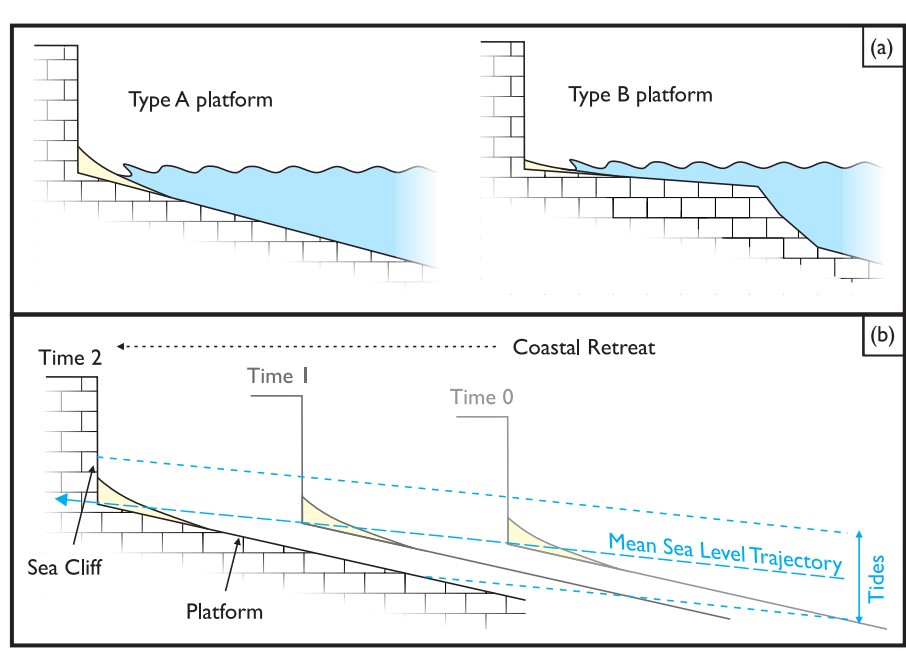

**Figure 2.** (a) End-member types of shore platform as defined by Sunamura (1992). (b) Illustration of steady state shore profile retreat subject to relative sea level rise.





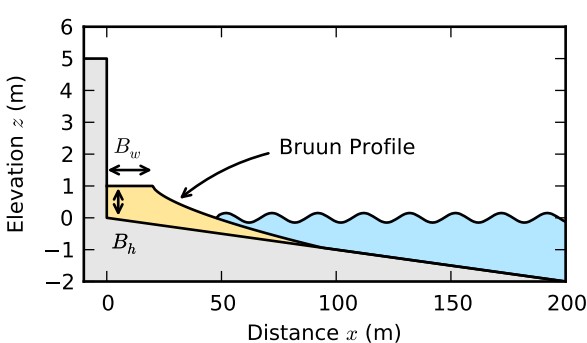

**Figure 3.** Example of beach model used in modelling $^{10}$Be accumulation in a shore platform. Beach is defined by a beach width $B_w$ and berm height $B_h$ and a Bruun profile (Bruun, 1954) seaward of the defined beach width.





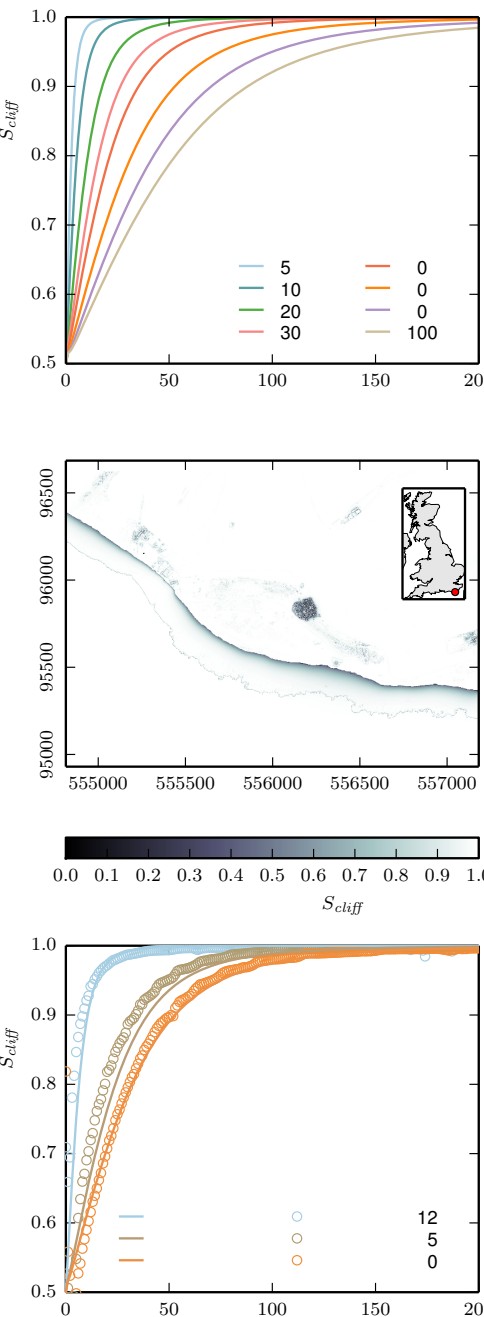

**Figure 4.** Topographic Shielding due to sea cliff. (a) Analytical model for topographic shielding on a shore platform as a function of distance from cliff measured according to Equation 11 for cliff heights ranging from 5 to 100 m. (b) Example map of the distribution of topographic shielding across a shore platform at Beachy Head, East Sussex, UK. (c) Comparison of Equation 11 to values measured using distributed shielding routines (Codilean, 2006; Mudd et al., 2016).





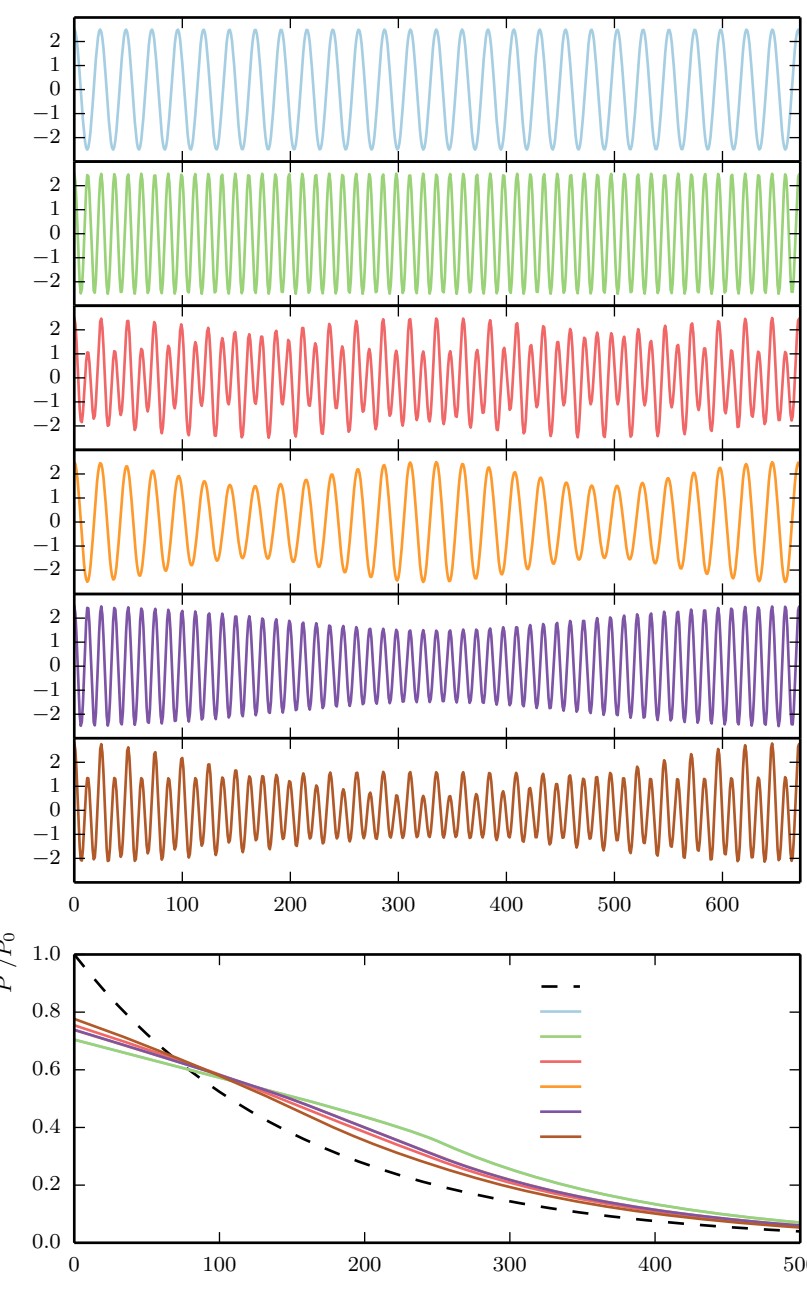

**Figure 5.** Influence of tidal regime on the distribution of $^{10}$Be Production across a shore platform. (a)-(f) Tidal water levels over a 28 day period for the hypothetical tidal regimes explored here (see Table 1). (g) Production rates relative to the case where there is no water shielding ($h_w = 0$) as a function of distance offshore on a shore platform with gradient $\alpha = 1/100$ for the different tidal regimes plotted in (b)-(g).

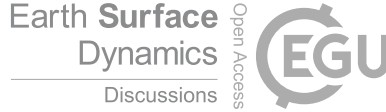



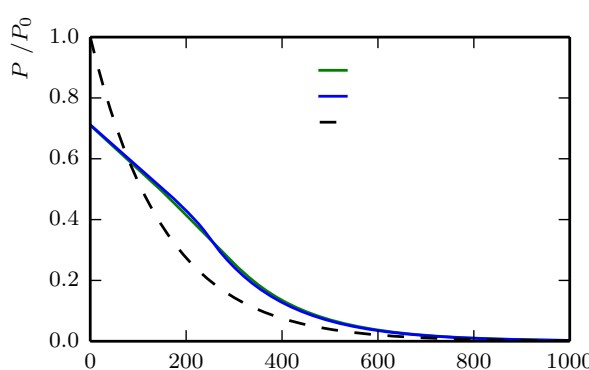

**Figure 6.** Distribution of $^{10}$Be Production across a shore platform with gradient $\alpha = 1/100$ for a simple diurnal tide model with amplitude 2.4 m and for real tide data from Newhaven, East Sussex, UK. Note there is very little difference in production between the two, suggesting a diurnal model with representative amplitude is sufficient for the purpose of modelling $^{10}$Be production.





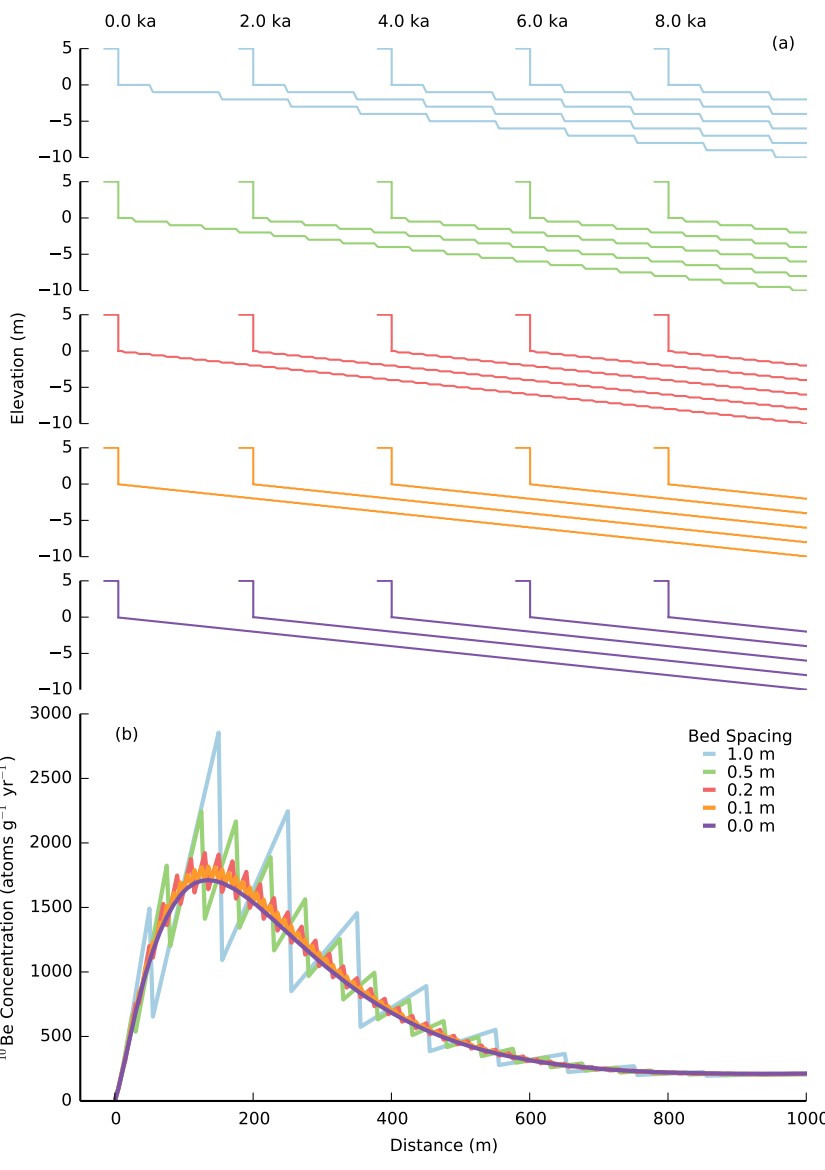

**Figure 7.** (a) Evolution of stepped shore platform profiles over 10 ka for step sizes (bedding thickness) up to 1 m. (b) Corresponding distribution of [10]Be as a function of step size. Large steps due to block removal processes result in a saw-toothed distribution superimposed on the expected humped distribution. Variability in the platform surface concentration increases with step size.





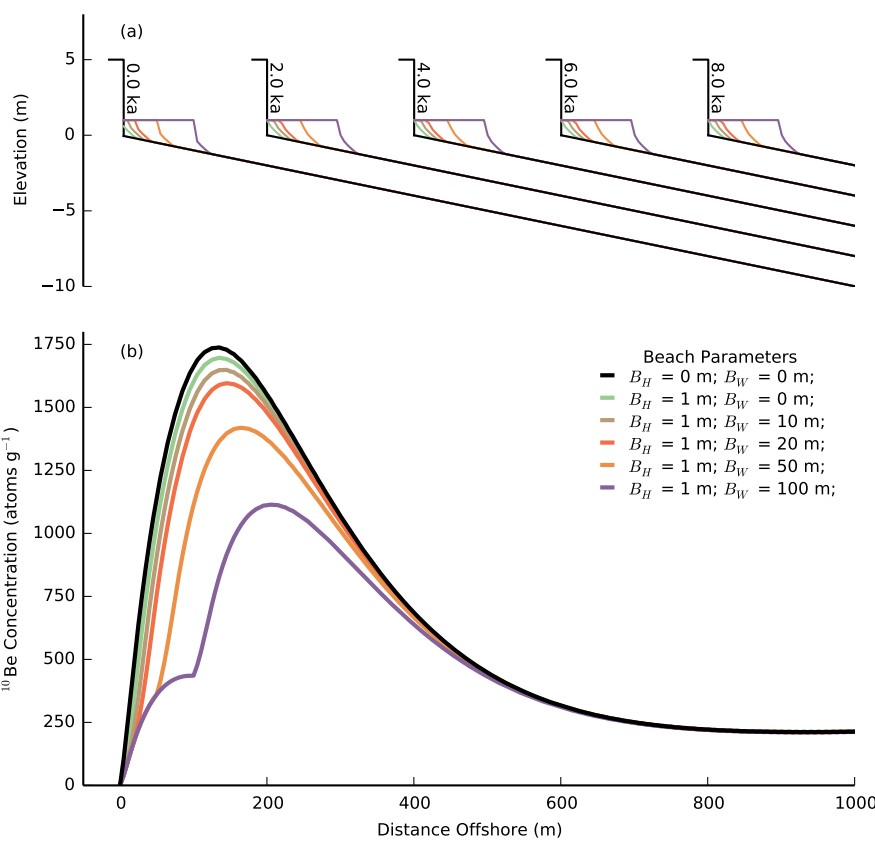

**Figure 8.** (a) Evolution of shore platform profiles over 10 ka with constant beach cover in the nearshore described by a Bruun profile (Equation 6). (b) Corresponding distribution of $^{10}$Be as a function of beach width. Beach cover reduces the concentrations of $^{10}$Be in the platform surface, and wider beaches result in lower $^{10}$Be concentrations and cause the position of the peak in concentrations to be further out from the cliff. Note that quite extreme beach cover $B_W \geqslant 50$ m is required to significantly ($\geqslant 15$ %) reduce platform $^{10}$Be concentrations.





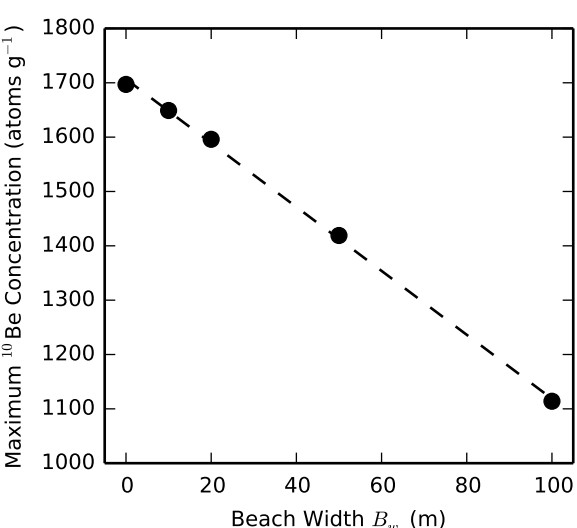

**Figure 9.** Reduction in the magnitude of peak concentration of $^{10}$Be with increasing beach width. Black symbols shows the peak concentrations taken from Figure 8. The relationship is approximately linear as shown by the dashed line fit by least squares regression. Note that quite extreme beach cover $B_W \geqslant 50$ m is required to significantly ($\geqslant 15\,\%$) reduce platform $^{10}$Be concentrations.





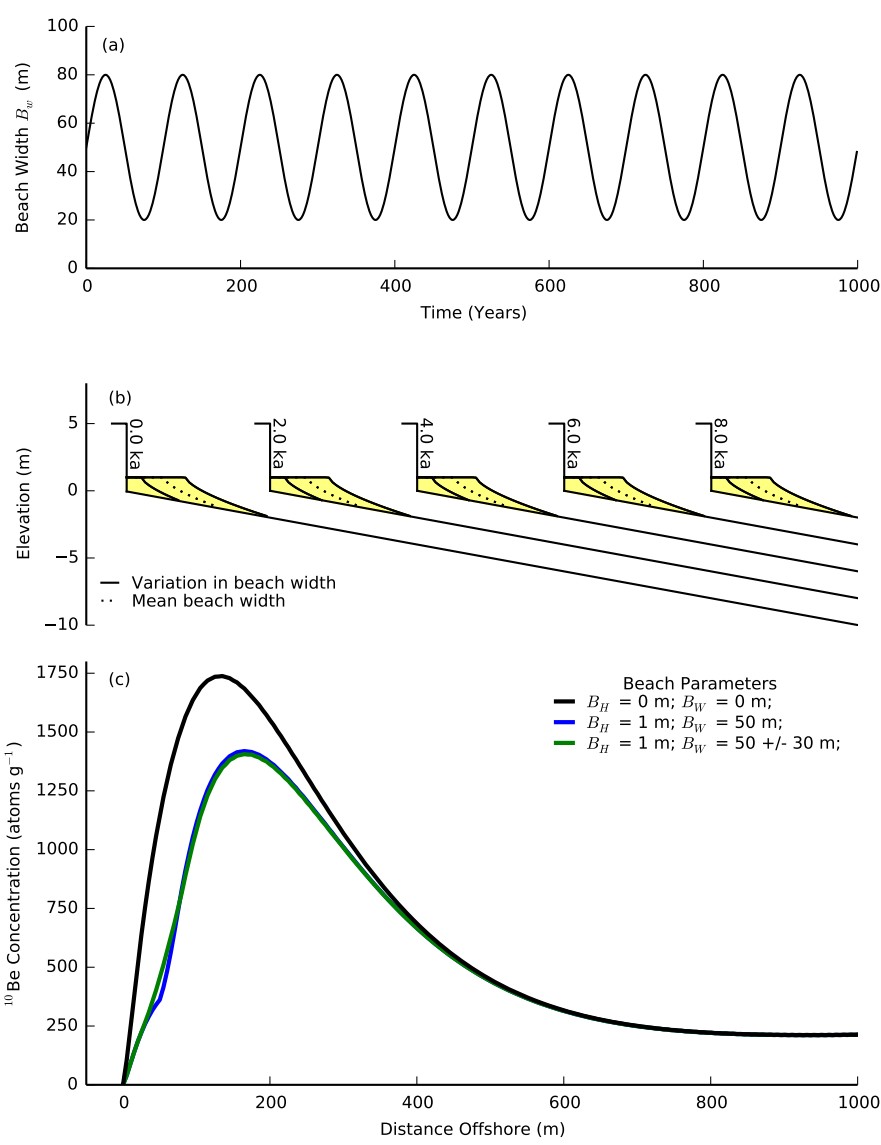

**Figure 10.** (a) Variation in beach width through time modelled as a sinusoidal function with a beach width $B_w = 50 \pm 30$ m and wavelength 100 years. (b) Evolution of shore platform profiles over 10 ka with variable beach cover in the nearshore described by a Bruun profile (Equation 6). (c) Corresponding distribution of $^{10}$Be for the case with no beach cover, fixed $B_w = 50$ m, and variable beach cover. Note that there is little difference in $^{10}$Be concentrations between a constant beach cover and variable beach cover.





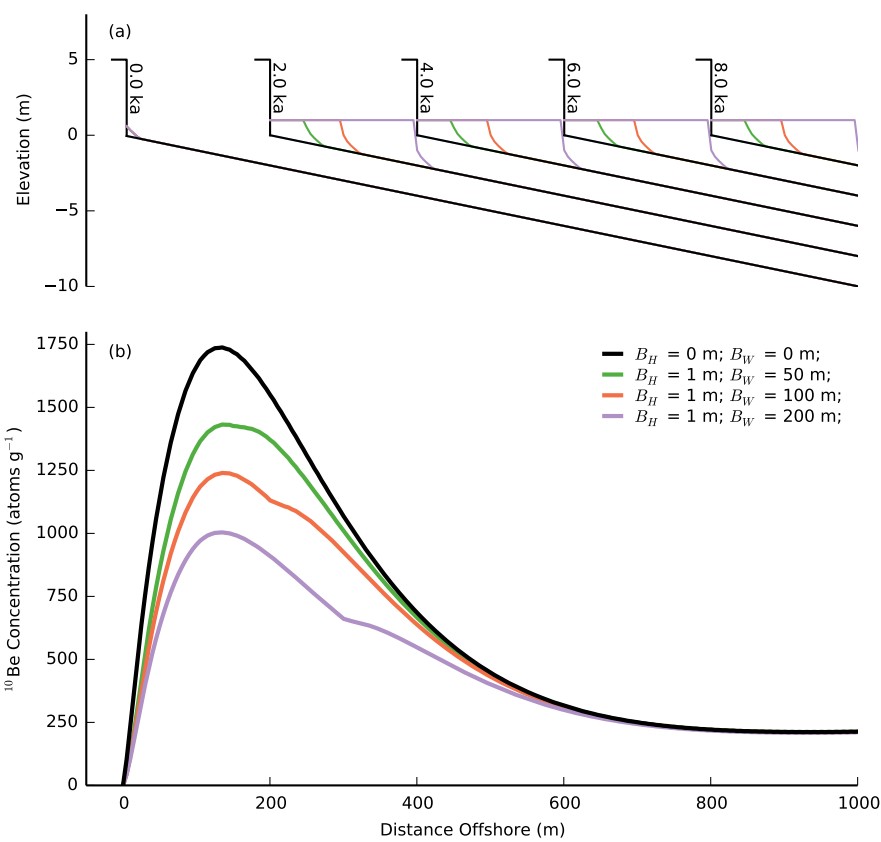

**Figure 11.** (a) Evolution of shore platform profiles over 10 ka with variable beach cover in the nearshore described by a Bruun profile (Equation 6). Beach width reduces to $B_w = 0$ m during the last 1000 yrs of the model runs. (b) Corresponding distribution of $^{10}$Be for the case with no beach cover and thinning beaches during the last 100 yrs. The presence of beaches that dissapeared over the last 1000 yrs would result in lower concentrations of $^{10}$Be in the platform surface.





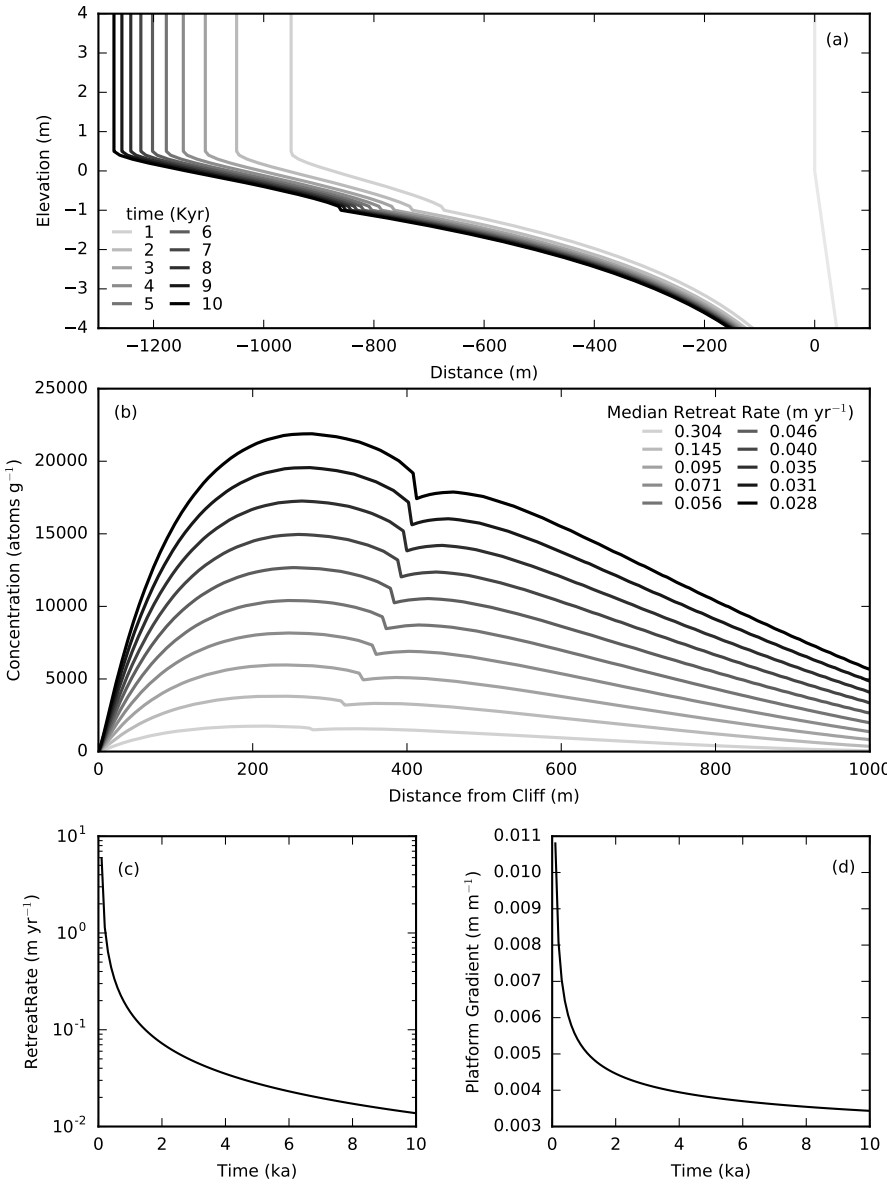

**Figure 12.** Transient simulations of shore profile evolution and $^{10}$Be concentrations for the case with no sea level rise. (a) Evolution of shore platform profiles over 10 ka driven by RoBoCoP (colour-coded in 1 ka interval). (b) Corresponding $^{10}$Be concentration predictions, normalised to the position of the cliff, (colour-coding also corresponds to the median retreat rates listed). Retreat rates are initially rapid and decline through time (c) and show a humped distribution, similar to steady state model predictions. The magnitude of the hump increases through time as the length of exposure increases and cliff retreat rates and platform downwear rates decline, resulting in a decrease in platform gradient through time (d).





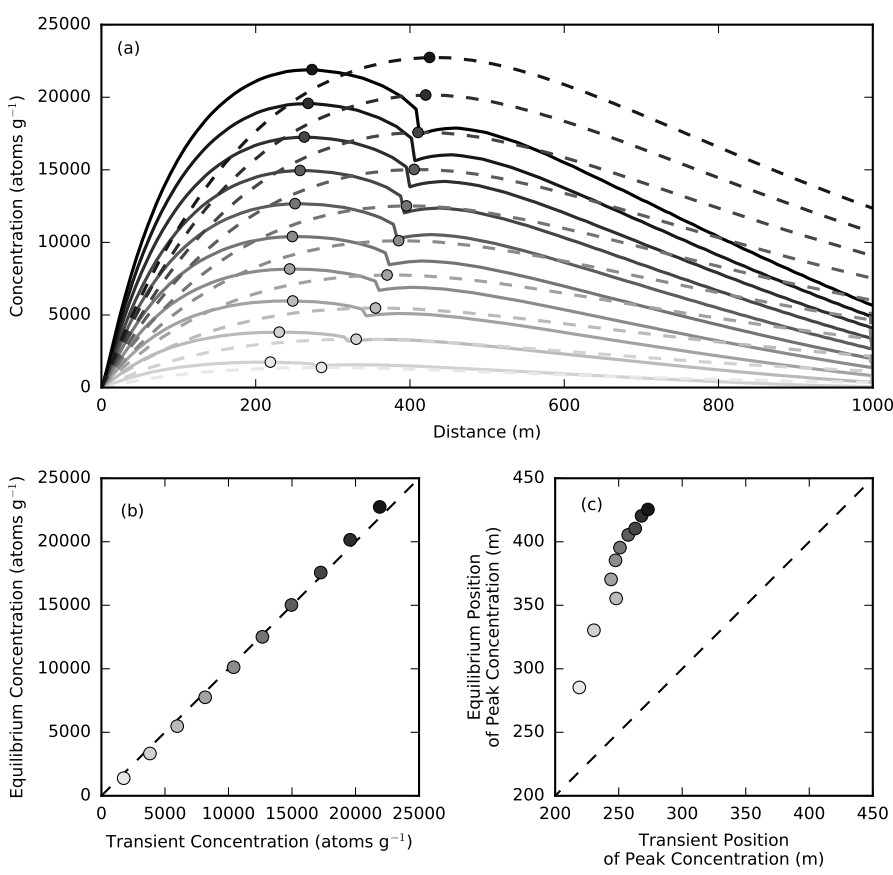

**Figure 13.** Comparison of transient simulations from Figure 12 to predictions assuming steady state profile retreat. Steady state models are run with constant retreat rate taken as the median values in the transient simulations (Figure 12). Color coding corresponds to the median retreat rates reported in Figure 12b, and the instantaneous mean platform gradient for each profile in 12a. (a) Distribution of $^{10}$Be concentration predictions. The magnitude of the hump is similar as shown in (b), but the location of the peak in concentration is further offshore when steady state retreat is assumed, as also shown in (c).





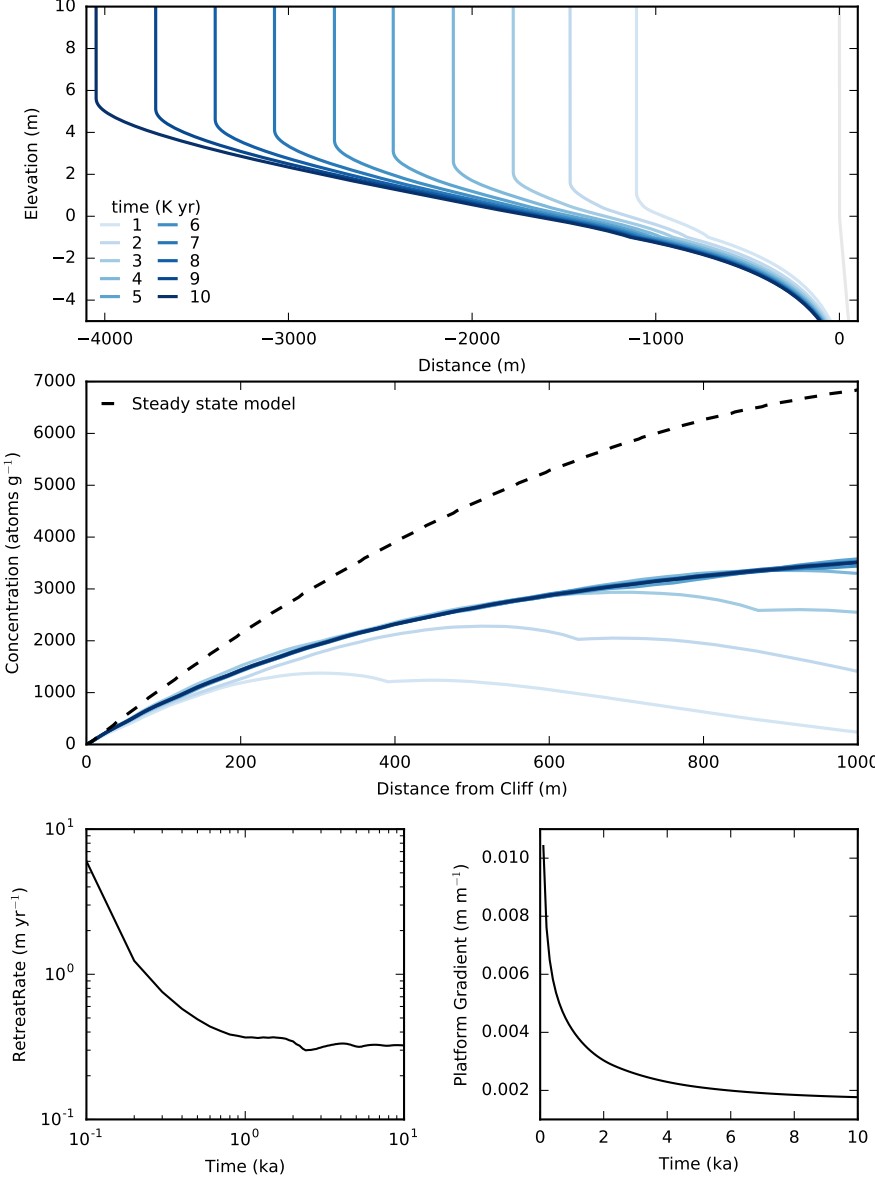

**Figure 14.** Transient simulations of shore profile evolution and $^{10}$Be concentrations for the case with 0.5 mm yr$^{-1}$ relative sea level rise. (a) Evolution of shore platform profiles over 10 ka driven by RoBoCoP (1 ka interval). Retreat rates are intially rapid but steady to 32 cm yr$^{-1}$ after 1 K yrs. Platform gradient declines through time. (b) Corresponding $^{10}$Be concentration predictions, normalised to the position of the cliff. Note that $^{10}$Be concentrations are consistent with distance from the cliff throughout the model runs, consistent with steady-state morphological retreat. The dashed line shows the predictions of a steady-state morphological model with retreat rate of 32 cm yr$^{-1}$, platform gradient of 1/500 and relative sea level rise rate of 0.5 mm yr$^{-1}$. Steady state model predicts higher concentrations than transient simulations by about a factor of 2.



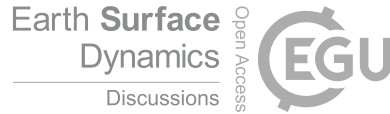

**Table 1.** Tidal constituents used to generate synthetic tides to explore variation in [10]Be production under different tidal regimes.

| Tidal | Tide A | | Tide B | | Tide C | | Tide D | | Tide E | | Tide F | |
| Constituent | $H_n$ | $\sigma_n$ | $H_n$ | $\sigma_n$ | $H_n$ | $\sigma_n$ | $H_n$ | $\sigma_n$ | $H_n$ | $\sigma_n$ | $H_n$ | $\sigma_n$ |
|---|---|---|---|---|---|---|---|---|---|---|---|---|
| $K1$ | 2.5 | 15.041 | - | - | 0.7 | 15.041 | 2.0 | 15.041 | - | - | 0.4 | 15.041 |
| $K2$ | - | - | - | - | - | - | - | - | - | - | 0.1 | 30.082 |
| $M1$ | - | - | - | - | - | - | - | - | - | - | 0.1 | 14.492 |
| $M2$ | - | - | 2.5 | 28.984 | 1.8 | 28.984 | - | - | 2.0 | 28.984 | 1.5 | 28.984 |
| $N2$ | - | - | - | - | - | - | - | - | 0.5 | 28.440 | 0.5 | 28.440 |
| $O1$ | - | - | - | - | - | - | 0.5 | 13.943 | - | - | 0.2 | 13.043 |



**Table 2.** Global parameters used in modelling $^{10}$Be accumulation in a shore platform

| Description | Symbol | Value |
|---|---|---|
| Acceleration due to Gravity | $g$ | 9.81 m s$^{-2}$ |
| Density of sea water | $\rho_w$ | 1025 kg m$^{-3}$ |
| Density of rock | $\rho_r$ | 1800 kg m$^{-3}$ |
| Surface production rate (spallogenic) | $P_0(s)$ | 4.0 atoms g$^{-1}$ yr$^{-1}$ |
| Surface production rate (muogenic) | $P_0(\mu)$ | 0.028 atoms g$^{-1}$ yr$^{-1}$ |
| $^{10}$Be decay constant | $\lambda$ | $4.99 \times 10^{-7}$ |
| Attenuation rate (spallogenic) | $\Lambda_s$ | 1600 kg m$^{-2}$ |
| Attenuation rate (muogenic) | $\Lambda_\mu$ | 42000 kg m$^{-2}$ |

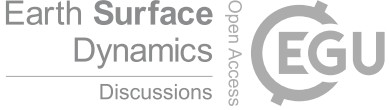

**Table 3.** Parameters used in modelling transient shore platform evolution

| Description | Symbol | Value |
|---|---|---|
| Initial platform gradient | $\alpha$ | 0.1 m m$^{-1}$ |
| Offshore Wave Height | $H_0$ | 1.0 m |
| Wave Period | $T$ | 6 s |
| Resistance Coefficient | $K$ | $10^{-4}$ m s kg$^{-1}$ |
| Wave energy dissipation coefficient | $k$ | 0.2 (Dimensionless) |
| Tidal amplitude | $H_n$ | 1.0 m |
| Angular speed | $\sigma_n$ | 28.984 ° hour$^{-1}$ |