# Peer review of "Controls on the distribution of cosmogenic $^{10}\text{Be}$ across shore platforms"

_Earth Surface Dynamics, 2016_

## Short Comment (SC1) · 10 Aug 2016

I have read this manuscript with strong interest as it can be viewed as a natural continuation of the paper we published in 2012. Many of the complications we thought to are here nicely addressed and I totally agree with the methods and conclusions of the paper. My only regret is that even if they focus on Sussex for tides and RSL there is no 10Be data from a natural shore platform. I am looking forward reading such data in a future paper.

I have just a couple of minor remarks.

The parameters of the beach cover are not explicit. How does a beach affect cosmogenic nuclide enrichment? Do you take a typical porosity and grain density? Is this shielding effect modulated by tides?

[Figure]

In part 6.2 the authors plead for mapping downwearing rates over the platform. This is challenging but I think it is currently a very hard work. Another possibility could be through LiDAR scanning, but the time span between two campaigns must be sufficiently long to overcome the low downwearing rates. In Mesnil Val we acquired such a picture in 2008 and we wait... Other attempts of interest are the evaluation of block removal: this can give the typical thickness and the importance of the phenomenon (backwearing) on the total downwearing (see Dornbusch and Robinson 2011 and Regard et al. 2013).

Dornbusch, U., and Robinson, D.A., 2011, Block removal and step backwearing as erosion processes on rock shore platforms: a preliminary case study of the chalk shore platforms of south-east England: Earth Surface Processes and Landforms, v. 36, p. 661–671, doi: 10.1002/esp.2086. Regard, V., Dewez, T.J.B., Cnudde, C., and Hourizadeh, N., 2013, Coastal chalk platform downwearing modulated by step backwearing and debris shielding: example from Normandy and Picardy (northern France): In: Conley, D.C., Masselink, G., Russell, P.E. and O'Hare, T.J. (eds.), Proceedings 12th International Coastal Symposium (Plymouth, England), Journal of Coastal Research, v. Special Issue No. 65, p. 1692–1697.

---

## Author Comment (AC1) · 10 Aug 2016

We would like to thank Vincent Regard for his encouraging comments. We have collected $^{10}$Be measurements from shore platforms in East Sussex which will be the focus of a future publication.

We are grateful for these remarks and will clarify the details of the influence of beach cover on $^{10}$Be production if invited to revise the manuscript. The average production rate at the beach surface is calculated as the average production rate across a single tidal cycle, the same method as in the commenter's own work (Regard et al., 2012). The production rate decays exponentially with beach depth following Equation 8 in our manuscript. Critically, the beach material is assumed to have the same density as bedrock, and we do not adjust this density based on wetting and drying by the tides as

the commenter is suggesting. Future site specific studies should be concerned about these nuances. Our exploratory modelling, however, was intended to highlight the first-order sensitivity of platform $^{10}$Be concentrations to the presence and variation in beach cover.

Sections 6.1 to 6.3 were intended, in part, to highlight some of these outstanding challenges that arise following Regard's own work and the experiments we have performed here. We cited the block removal study from south-east England (Dornbusch and Robinson, 2011) but were unaware of the study on the French coast (Regard et al., 2013), which we will also cite in any revision. In particular, and as discussed in the manuscript, Figure 7 reveals the potential to document rates of step back-wearing, sampling the shore platform at high density. The two sites highlighted in this comment would be excellent places to test such an approach. With regards to platform down-wearing, our modelling results suggest there is need to couple $^{10}$Be measurements with observations of platform downwearing rates. Sites where downwearing has been observed using micro-erosion-metres might be appropriate because the records are somewhat longer than those available through topographic surveying (LiDAR + multi-beam bathymetry) at this stage. But as the commenter suggests, these efforts will become more fruitful as time elapse.

References:

Dornbusch, U., and Robinson, D.A., 2011, Block removal and step backwearing as erosion processes on rock shore platforms: A preliminary case study of the chalk shore platforms of south-east England: Earth Surface Processes and Landforms, v. 36, no. 5, p. 661–671, doi: 10.1002/esp.2086.

Regard, V., Dewez, T., Bourlais, D.L., Anderson, R.S., Duperret, A., Costa, S., Leanni, L., Lasseur, E., Pedoja, K., and Maillet, G.M., 2012, Late Holocene sea-cliff retreat recorded by 10Be profiles across a coastal platform: Theory and example from the English Channel: Quaternary Geochronology, v. 11, p. 87–97, doi:

10.1016/j.quageo.2012.02.027.

Regard, V., Dewez, T.J.B., Cnudde, C., and Hourizadeh, N., 2013, Coastal chalk platform erosion modulated by step erosion and debris shielding: example from Normandy and Picardy (northern France): Journal of Coastal Research, v. 12th Inter, no. 65, p. 1692–1697, doi: 10.2112/SI65-286.1.

---

## Referee Comment (RC1) · A. Trenhaile (Referee) · 25 Aug 2016

Cosmogenic dating was initially greeted by rock coast workers, as a quasi-panacea, a long awaited solution to the question, that had arisen over more than a century, regarding the age of shore platforms, and especially whether they are primarily contemporary (Holocene) or inherited (from previous interglacials when sea level was similar to today's) landforms. Related to this question and of more immediate practical concern was its apparent ability to provide reliable data on rates of cliff recession over lengthy periods during the Holocene, and to test predictive models and better predict the effects of climate change, and especially rising sea level. There have been a few useful 10Be dating applications. They include the work of Choi et al. (2012), who suggested that the seaward portions of shore platforms in Korea are up to about 150 thousand years old, and that the modern platform is cutting into its interglacial predecessor. Conversely,

[Figure]

Regard et al., (2012) opined that in northern France, the mean rate of cliff retreat since the mid-Holocene has been 11–13 cm yr-1, which would have been sufficient to create 'contemporary' intertidal shore platforms hundreds of metres in width. Despite these valuable contributions, I would contend that cosmogenic nuclide analysis, whether for dating landforms or quantifying their rates of erosion has failed, as yet, to revolutionize the study of rock coasts. Given this, the present paper, which discusses and models the effect of several of the factors and assumptions that constrain the use of cosmogenic analysis on rock coasts, represents a welcome addition to the literature and one which will, no doubt, improve our ability to understand the longterm development of rock coasts. Nevertheless, this paper once again points out the inherent limitations of cosmogenic analysis on rocky coasts for which we have little longterm data on such factors as downwearing and backwearing rates (erosion in the vertical and horizontal planes, respectively) and historical and short-term (storm and calm condition) variations in sediment thickness, type, and extent.

My main criticism of this paper is the almost complete neglect of downwearing by wearing processes, including by tidal wetting and drying and salt weathering. The paper notes that models have indicated that shore platforms and cliffs trend towards a morphological steady state. There are certainly many examples of steady states in model predictions with constant sea level, although when one considers the effect of changing relative sea level, simulated longterm platform development is more complex (see for example: Trenhaile, A. S. 2001: Modeling the Quaternary evolution of shore platforms and erosional continental shelves . Earth Surface Processes and Landforms 26, pp. 1103-28, and Trenhaile, A. S. 2014. Modelling the effect of Pliocene-Quaternary changes in sea level on stable and tectonically active land masses. Earth Surface Processes and Landforms 39, 1221-35). The paper quotes our work (Porter et al., 2010) in eastern Canada (on page 15 of the manuscript), but essentially ignores the relevant conclusions that relate to spatial variations across the intertidal zone. This paper showed, based on laboratory experiments lasting several years and about 2000 rock samples, together with about 200 transverse micro-erosion metre stations in the field,

that, while weathering downwearing rates (isolated in the laboratory experiments) tend to be a maximum in the upper intertidal zone, there is no clear pattern in the field, possibly because of the effect of other erosional mechanisms with different elevational efficiencies. Whether there are weathering patterns or not, mean rates of downwearing recorded by numerous workers in different environments and on different types of rock, which generally range from almost 0 up to a few millimeters per year (and hence significant lowering per millennium), surely have an important effect on predicted erosion rates. The reliance on models of platform development (whether parallel or declining slope retreat) to represent rates of surface lowering (as opposed to field measurements and micro-erosion metre data) remains an inherent weakness in attempts to apply cosmogenic techniques to rocky coasts.

Minor points include:

The traditional distinction (classification) between sloping and subhorizontal (or to use Sunamura's terminology, type A and B platforms) belies the fact that there is a continuous spectrum of forms with gradients that reflect tidal range and rock hardness (resistance), as well as possibly sediment grain size and the effect of Holocene sea levels higher that today .

It is not clear to me why, on line 12 or page 4, abrasion is assumed to be dominant with a steady state that assumes platform downwearing is gradual and constant (especially since weathering is not considered). Sediment on shore platforms accumulates preferentially at the cliff foot and in structural depressions on the platform. In the latter case, abrasion rates with vary enormously across the intertidal zone in an essentially random manner. Abrasion towards the rear of the platform is spread over the intertidal surface as the cliff retreats, but constant abrasion assumes constant wave and sea level conditions as well as constant beach types and amounts.

Page 4, line 20, many models do/did not emphasize erosion at the water surface but rather erosion due to wave generated bottom currents (which is only effective in clays

and other 'soft' rocks.

An alternative to the Bruun Rule, which has been criticized by several workers and in any case is not designed for beaches with rigid (shore platform) foundations, is given by: Trenhaile, A. S. 2004. Modeling the accumulation and dynamics of beaches on shore platforms. Marine Geology 206, 55-72.

The paper needs some proof-reading. There are other examples that could be given but note 'ditributed' on line 14 of page 10 and on line 15 it should be 'adjacent to the cliff'.

On lines 12 to 13 on page 11 it is claimed that there is no downwearing on top of flat-topped steps. This is wrong - weathering certainly lowers the surface until (and in some cases before) it is eliminated by step retreat.

Lines 10 to 13 on page 13 emphasize errors due to differences in predicted downwearing rates according to the two evolutionary models. The reality is likely to be worse owing to weathering induced downwearing (with spatial patterns, if any, that we do not yet understand).

Alan S. Trenhaile

---

## Referee Comment (RC2) · M. Dickson (Referee) · 26 Aug 2016

This is a welcome addition to a rather slowly emerging literature on the application of cosmogenic dating to inform rocky shore evolution studies. The proposed model usefully extends the work described by Regard et al. (2012) and includes I think the majority of the factors required. In doing so, the paper highlights that the use of this method to determine erosion rates on rocky shores, while conceptually simple, in practice is rather more complex. Nevertheless, the lack of quantitative measurements of rates of change has been a long-standing limitation in our field, so efforts in the direction of this paper are welcome.

Below are some comments that the authors might wish to consider during any revision. Please note that these comments integrate thoughts from Hironori Mat-

sumoto who is currently making further developments to a rocky shore profile evolution model (Matsumoto, H., Dickson, M. E., and Kench, P. S.: An exploratory numerical model of rocky shore profile evolution, Geomorphology, 268, 98–109, doi:10.1016/j.geomorph.2016.05.017, 2016.)

Can you clarify how the model considers platform slope? Does it calculate the cliff toe position with a fixed platform geometry, similar to previous tide-less models (Sunamura, 1975)? Perhaps not, because later, in section 5.5, there is reference to the gradient of the shore platform decreasing through time, as the platform widens. There is also reference to the emergence of a 'stepped' platform - what is the reason for this? Ultimately it is a little unclear how exactly the geometry is handled. The text states that the model is similar to the existing models of Trenhaile (2000), Walkden and hall (2005), and Matsumoto et al (2016), which all consider tide and vertical components. It would be useful to expand and clarify similarities and differences in that regard.

Did the authors consider prospects for using the model to test/discuss factors that affect 10Be concentrations on other types of shore platform geometry beyond the sloping (type-A) platform investigated? The paper refers to type-B platforms, raising the question as to whether the model adds any new knowledge of likely differences in concentrations across different possible platform geometries.

The model considers topographic shielding in the case of a constant cliff height. A further exploration of interest would be to consider a slowly increasing cliff height. What comes to mind is the example of progressive cliffing into hillslopes rounded over long glacial periods. This could introduce further complications for 10Be concentrations, because erosion into cliffs of increasing height might progressively increase beach thickness (5.4.1)... assuming no gradient in alongshore sediment flux. The number of potential model scenarios can quickly increase, but it would be interesting to have a somewhat expanded discussion of this factor.

There is some interesting discussion on the implications of platform downwear, but I

agree with the other reviewer that this aspect of the modelling could be improved - he makes a number of points to consider in that regard which I have not expanded on here.

In concluding, is it possible to tabulate or summarise somehow the relative sensitivities of the different factors?

Minor points: - Please double check that you have the exponent written correctly in equation 5 (i.e. '+' hw(t))? - Please double check the wording of the last sentence on p7 (upper / lower?) - two "and"s p4 line 28 - Clarify what is meant by significant (para 8, line 16) - Mis-spelling of fund[e]mentally - Grammar problem para 9 line 24 - no full stop before Failure p15 l12 - Choi2012 p16

---

## Author Comment (AC2) · 6 Oct 2016

**Response to review**

We would like to thank Dr. Alan Trenhaile for his positive and constructive comments. Below are our responses to all of these comments including details of any changes we have made to the manuscript guided by the reviewer's suggestions. We have numbered the reviewer comments and they are coloured black, while our responses are coloured blue, and italicised where we are quoting directly from the revised manuscript.

1. Cosmogenic dating was initially greeted by rock coast workers, as a quasi-panacea, a long awaited solution to the question, that had arisen over more than

a century, regarding the age of shore platforms, and especially whether they are primarily contemporary (Holocene) or inherited (from previous interglacials when sea level was similar to today's) landforms. Related to this question and of more immediate practical concern was its apparent ability to provide reliable data on rates of cliff recession over lengthy periods during the Holocene, and to test predictive models and better predict the effects of climate change, and especially rising sea level. There have been a few useful [10]Be dating applications. They include the work of Choi et al. (2012), who suggested that the seaward portions of shore platforms in Korea are up to about 150 thousand years old, and that the modern platform is cutting into its interglacial predecessor. Conversely, Regard et al., (2012) opined that in northern France, the mean rate of cliff retreat since the mid-Holocene has been 11–13 cm yr$^{-1}$, which would have been sufficient to create 'contemporary' intertidal shore platforms hundreds of metres in width. Despite these valuable contributions, I would contend that cosmogenic nuclide analysis, whether for dating landforms or quantifying their rates of erosion has failed, as yet, to revolutionize the study of rock coasts.

The two studies discussed by the reviewer above (plus another nice paper by Rogers et al. (2012)) have done an excellent job in demonstrating that there is significant potential for cosmogenic radionuclides (CRNs) to yield evidence for long-term processes and rates in rocky coast settings. As the reviewer points out, measurement of cosmogenic radionuclides has not been able to "revolutionise the study of rock coasts", nor would we expect it to. Rather, they provide an additional tool in the coastal geomorphologist's repertoire to help understand the rates and processes by which cliffed coasts and shore platforms have evolved in the past, and notably, over timescales longer than those at which observational data has been collected. We were motivated by reviewer's own conclusions when reviewing climate change impacts on rocky coasts that "... we must... acquire more precise and longer records of erosion rates and other expressions of process efficacy and their relationship to prevailing morphogenic conditions" (Trenhaile, 2014, p. 15). It remains to be seen whether cosmogenic radionuclides can improve the precision of erosion rate estimates, but it certainly has the potential to provide longer records (over centennial to millennial timescales) in erosive environments in which there is no alternative evidence of a previous state.

2. Given this, the present paper, which discusses and models the effect of several of the factors and assumptions that constrain the use of cosmogenic analysis on rock coasts, represents a welcome addition to the literature and one which will, no doubt, improve our ability to understand the longterm development of rock coasts. Nevertheless, this paper once again points out the inherent limitations of cosmogenic analysis on rocky coasts for which we have little longterm data on such factors as downwearing and backwearing rates (erosion in the vertical and horizontal planes, respectively) and historical and short-term (storm and calm condition) variations in sediment thickness, type, and extent.

We are pleased that the reviewer finds our work to be "a welcome addition to the literature" and we hope that it can "improve our ability to understand the long-term development of rock coasts" through the application of CRNs in such environments. We felt that it was a critical step to further explore assumptions that may be required in order for future studies to interpret concentrations of CRNs in terms of process rates on shore platforms so that sampling strategies might be better informed to test these assumptions. This is particularly important given the expense of making CRN measurements that may limit the number of samples that can be acquired in a single study. As an example our results suggest that there is the potential to estimate rates of block removal on stepped platforms through high density sampling and this would be an exciting avenue for future work. We have not made any modifications to the paper based on these opening statements as we feel the context and our aims are already set out clearly in the manuscript, if the reviewer had any specific modifications to our introductory discussion in mind then we would welcome further comment.

3. My main criticism of this paper is the almost complete neglect of downwearing by wearing processes, including by tidal wetting and drying and salt weathering.

We acknowledge this criticism and its importance, and are not suggesting that these processes are not important mechanisms of downwear on shore platforms. Previous studies that have tried to quantify cliff retreat rates have relied on the assumption of steady-state retreat whereby a constant shore platform morphology is translated landward through time. In this study we have applied a dynamic morphological evolution model, guided by the reviewer's own work (e.g. Trenhaile, 2000), that is capable of generating the sorts of behaviour we wished to explore (e.g. shore platforms that widen with time). We have thus gone further than any previous studies of cosmogenic radionuclides in coastal settings when considering the morphological evolution of the coast. All numerical models are abstractions of the real world at some level but our experiments allow us to explore a number of interesting behaviours such as both continuous and discontinuous distribution of erosion (gradual downwear vs block removal). Such a model, at first order, broadly recreates the patterns of downwear observed from short-term monitoring data, with downwear decreasing with decreasing elevation on the shore platform (see also response to reviewer comment #5).

4. The paper notes that models have indicated that shore platforms and cliffs trend towards a morphological steady state. There are certainly many examples of steady states in model predictions with constant sea level, although when one considers the effect of changing relative sea level, simulated longterm platform development is more complex (see for example: Trenhaile, A. S. 2001: Modeling the Quaternary evolution of shore platforms and erosional continental shelves . Earth Surface Processes and Landforms 26, pp. 1103-28, and Trenhaile, A. S. 2014. Modelling the effect of Pliocene-Quaternary changes in sea level on stable and tectonically active land masses. Earth Surface Processes and Landforms 39, 1221-35).

In this paper we do not explore the history of shore platform development throughout the quaternary but rather focus on the development of shore platforms during the Holocene (more specifically, the last 7-8 thousand years, the time over which eustatic sea level has been relatively stable). Where shore platforms are inherited features formed perhaps during a previous interglacial period, subsequent exposure would result in CRN concentrations substantially higher than those from contemporary platforms, as demonstrated by Choi et al. (2012). We have clarified this in the introduction by adding a more detailed description of the findings of Choi et al. (2012) and by clarifying that we are only concerned in this study with Holocene platforms stating:

*"Some shore platforms may have formed during the Holocene and thus be entirely contemporaneous features, whilst others, particularly wide platforms in resistant lithologies, have been interpreted as inherited features, formed during previous sea level high stands and reoccupied during the Holocene (e.g. Blanco Chao et al., 2003). Evidence for the antiquity of shore platforms may be revealed by CRNs. For example, Choi et al. (2012) measured $^{10}$Be concentrations in shore platforms cut into resistant lithologies on the Korean coast. High concentrations were consistent with modelled ages extending back as far as 142 ka, and importantly, these ages did not correct for weathering and erosion and thus should be considered as minimum ages. Thus Choi et al. (2012) conclude that these platforms are at least partially inherited features, originated in the Pleistocene."*

*"In this study we quantified the sensitivity of platform CRN concentrations to topographic shielding, various processes of platform erosion/downwear, the presence/absence of beach cover, and transience in shore profile evolution. In doing so we consider only the development of shore platforms during the Holocene, over the timescale during which eustatic sea level has been relatively stable (7 ka to present). We addressed this with a numerical model coupling cross-shore coastal evolution and $^{10}$Be production to explore the potential for quantifying*

*coastal retreat rates from $^{10}$Be concentration measurements."*

5. The paper quotes our work (Porter et al., 2010) in eastern Canada (on page 15 of the manuscript), but essentially ignores the relevant conclusions that relate to spatial variations across the intertidal zone. This paper showed, based on laboratory experiments lasting several years and about 2000 rock samples, together with about 200 transverse micro-erosion metre stations in the field, that, while weathering downwearing rates (isolated in the laboratory experiments) tend to be a maximum in the upper intertidal zone, there is no clear pattern in the field, possibly because of the effect of other erosional mechanisms with different elevational efficiencies.

We are grateful to the reviewer for highlighting their findings, and apologise that we did not do justice to this excellent piece of work that explores weathering and erosion processes on shore platforms both in the field and through controlled experiments. The lack of a "clear pattern in the field" meant that we were not able to use the data from their study to parameterise our model, however the model does reproduce the general trend they observed, that downwearing tends to be highest in the upper intertidal zone, declining offshore. Our intention in this study has been to explore the general case, parsimoniously, to better understand the sensitivity of CRN concentrations to factors such as the potential for non-uniform distribution of downwearing, and our analyses are by no means exhaustive. Future work that measure CRN concentrations at specific sites where there are good observational datasets will be a logical and important next step.

That said, there is a distinct scaling issue between their experiments which take place over at most a 6 year period, and the centennial to millennial timeframe over which we are required to model coastal evolution in order to explore the controls on the build-up of $^{10}$Be. We do not feel it would be appropriate to take the results of a single site and extrapolate over several millennia for the purposes of this study, although site specific investigation using CRNs would be well advised to place their findings in the context of such observations. Nevertheless, Stephenson et al. (2012) found that rates of platform erosion averaged over a 32 year period were not significantly different from the two year record initially analysed at the same sites, concluding that low magnitude, high frequency events drive platform lowering at that site. It is not yet clear whether such short-term downwear rates can be extrapolated at other sites, in other lithologies, for different tidal regimes and different erosion processes. Ongoing work with micro erosion metres, combined with our growing ability to derive high resolution 3D surface models with LiDAR and photogrammetry will help to address this matter further and we would welcome such research.

Given all this, we have not carried out any additional experiments in light of the reviewer's criticism. If the reviewer has any specific requirements he would like to see then we would welcome them. Rather, we have expanded our discussion of these issues in the paper to highlight that there is still plenty to be done as detailed below.

At the end of the introduction we have added:

*"In this study we quantified the sensitivity of platform CRN concentrations to topographic shielding, various processes of platform erosion/downwear, the presence/absence of beach cover, and transience in shore profile evolution. Our intention is to explore the general case parsimoniously to better understand controls on $^{10}$Be concentrations, rather than to make accurate predictions for any specific field site."*

In justifying the development of a dynamic morphological model we now state:

*"Moreover, micro-erosion meter measurements of platform downwear suggest that downwear is not uniform across the shore profile, but tends to be faster in the upper intertidal zone and decline with depth (Porter et al., 2010), and to explore the influence of such a distribution on $^{10}$Be concentrations, a dynamic morphological model was required."*
In the discussion we have expanded our brief mention of downwear measurements to say:

*"Accounting for the distribution of downwear would be aided by data on the distribution of downwear rates from micro-erosion-meter measurements (e.g. Robinson, 1977; Porter et al., 2010) and there is evidence that rates measured over the short term (1-2 years) are consistent with rates measured over three decades (Stephenson et al., 2010, 2012). Nevertheless, it is still not clear whether it is appropriate to extrapolate these rates over centennial to millennial timescales required to accumulate measureable concentrations of $^{10}$Be, nor is it yet clear whether this result is consistent across different shore platforms around the world. Future studies that measure the distribution of CRN concentrations (both across the shore platform and at depth) at sites with long records of observed downwearing rates would be an important next step in this line of enquiry."*

6. Whether there are weathering patterns or not, mean rates of downwearing recorded by numerous workers in different environments and on different types of rock, which generally range from almost 0 up to a few millimeters per year (and hence significant lowering per millennium), surely have an important effect on predicted erosion rates. The reliance on models of platform development (whether parallel or declining slope retreat) to represent rates of surface lowering (as opposed to field measurements and micro-erosion metre data) remains an inherent weakness in attempts to apply cosmogenic techniques to rocky coasts.

The reviewer has highlighted an interesting issue in the application of cosmogenic radionuclides to rocky coasts that will need to be addressed in future studies, namely the disconnect between short-term observations and long-term platform evolution, as we have already mentioned above. Our exploration of the sensitivity of CRN concentrations is not site specific and thus the use of spot measurements from a particular location, extrapolated back throughout most of the Holocene does not seem appropriate. It strikes us that, despite the findings of Stephenson

et al (2010; 2012) that rates observed over a few years are similar to those over a few decades for two field sites, the extrapolation of short-term measurements (3 decades at best) of shore platform erosion over millennial timescales is also inherently weak. Future work measuring [10]Be concentrations at sites with good long-term field records of surface lowering would be an excellent future extension of the application of CRNs to rock coasts, and we have called for this in the paper (see our response to reviewer comment #5). Moreover, our growing capacity to observe changes in platform morphology in a distributed way using terrestrial LiDAR and photogrammetry in order to identify individual erosion events will lead to better understanding of the efficacy and distribution of platform downwear processes, and we commend the reviewer for pioneering these sorts of investigations.

As the reviewer points out both here and in his final comment, we do not yet understand the spatial distribution of weathering on shore platforms.

7. The traditional distinction (classification) between sloping and subhorizontal (or to use Sunamura's terminology, type A and B platforms) belies the fact that there is a continuous spectrum of forms with gradients that reflect tidal range and rock hardness (resistance), as well as possibly sediment grain size and the effect of Holocene sea levels higher that today.

We thank the reviewer for highlighting that such a classification is an over simplification and we agree that these two forms are effectively end-members of a much wider array of shore platform morphologies. Our experiments were intended to explore such intermediate platform morphologies, we do not limit ourselves to type-A and type-B platforms. However the beginning of section 2 may have been misleading on this point. We now begin this section by saying:

*"Cliffed, rocky coasts, are commonly fronted by shore platforms that have previously been classified into two types (Sunamura, 1992). Type-A platforms are characterised by a gently sloping erosional platform surface extending offshore*

*beyond maximum low water. Type-B platforms are shallow gradient to sub-horizontal and terminate at their seaward edge at maximum low water through a scarp (Figure 2a). Numerical models of shore platform evolution have successfully recreated both of these end-member morphologies, but have revealed that there are a range of other possible morphologies in between, for example sloping platforms terminating terminating in a scarp (e.g. Trenhaile, 2000; Walkden and Hall, 2005; Matsumoto et al., 2016)."*

8. It is not clear to me why, on line 12 or page 4, abrasion is assumed to be dominant with a steady state that assumes platform downwearing is gradual and constant (especially since weathering is not considered). Sediment on shore platforms accumulates preferentially at the cliff foot and in structural depressions on the platform. In the latter case, abrasion rates with vary enormously across the intertidal zone in an essentially random manner. Abrasion towards the rear of the platform is spread over the intertidal surface as the cliff retreats, but constant abrasion assumes constant wave and sea level conditions as well as constant beach types and amounts.

We thank the reviewer for highlighting the overly simple nature of this explanation. Our assumption is that in the case of steady-state retreat that platform lowering is constant and gradual across the shore platform. Over centennial to millennial timescales we might reasonably expect that the sorts of processes the reviewer describes here will tend to occur uniformly over a platform. Indeed, the existence of generally flat or planar platforms in some settings suggest this must be the case. We now state:

*"A steady state approach assumes platform downwear is gradual and constant across the entire profile, and proportional to the rate of cliff retreat. We are therefore assuming that the combined mechanical and chemical processes that can cause shore platform lowering culminate to constant and gradual downwear in this case."*

9. Page 4, line 20, many models do/did not emphasize erosion at the water surface but rather erosion due to wave generated bottom currents (which is only effective in clays and other 'soft' rocks.

We now state that: *"We developed a simple numerical model for shore profile evolution (the ROck and BOttom COastal Profile [RoBoCoP] Model), broadly similar to those of Sunamura (1992), Anderson et al. (1999) and Trenhaile (2000). These models assume..."*

10. An alternative to the Bruun Rule, which has been criticized by several workers and in any case is not designed for beaches with rigid (shore platform) foundations, is given by: Trenhaile, A. S. 2004. Modeling the accumulation and dynamics of beaches on shore platforms. Marine Geology 206, 55-72.

We thank the reviewer to drawing our attention to this alternative model for beach profile shape. We note that our model is not employing the criticised "Bruun Rule" that relates rates of shoreline retreat to relative sea level rise, but rather we use a "Bruun Profile" as a description of the shape of the beach profile. We chose a Bruun profile for its computational simplicity. Our goal was not to explicitly replicate beach behaviour but rather to explore the influence that shielding by sediment cover has on the accumulation of [10]Be in shore platforms. Thus we took a parsimonious approach to representing beach morphology. We note that Trenhaile (2004) takes a similarly simplistic approach, representing the shoreface as a planar slope of constant gradient, though that gradient varies depending on wave conditions and grain size.

11. The paper needs some proof-reading. There are other examples that could be given but note 'ditributed' on line 14 of page 10 and on line 15 it should be 'adjacent to the cliff'.

We apologise for the sloppy presentation despite our endeavours to catch all typological errors, and thank the reviewer for highlighting any such mistakes when

they have occurred. We have corrected the blunders explicitly pointed out by all reviewers, and our redraft has undergone thorough proof reading by all authors.

12. On lines 12 to 13 on page 11 it is claimed that there is no downwearing on top of flat-topped steps. This is wrong - weathering certainly lowers the surface until (and in some cases before) it is eliminated by step retreat.

This is not a claim since we are referring specifically to the model simulations: "there is no vertical downwear in these simulations". Our simulations exploring block removal processes are exploratory and can be considered end-member scenarios. We sought to understand the influence that block removal processes might have on [10]Be concentrations at the surface. Were there some vertical downwear then we would expect the gradients in [10]Be concentrations across the flat-topped steps to be lower, i.e. concentrations would not increase as rapidly in an offshore direction. The result presented in this section demonstrates that block removal processes will cause deviations from the expected "hump" shaped distribution, but the general form of the hump still remains, and this is encouraging for potential future applications.

13. Lines 10 to 13 on page 13 emphasize errors due to differences in predicted downwearing rates according to the two evolutionary models. The reality is likely to be worse owing to weathering induced downwearing (with spatial patterns, if any, that we do not yet understand).

Yes, the point we are trying to make is that the more rapid the rate of downwearing, the more [10]Be-laden rock is removed from the platform surface and the lower the subsequent production rate will be; therefore, the lower the CRN concentrations will be. This will lead to an over prediction of cliff retreat rates if not factored in (i.e. if one were to blindly assume steady-state retreat of a planar shore platform). If weathering processes are leading to more rapid downwear then this problem will be amplified. We have stressed in the discussion that measuring

CRN concentrations at sites with good records of downwear will be an important next step in developing our ability to estimate rates of cliff retreat, or platform downwear using CRNs. We have not added to the results section highlighted by the reviewer, but instead build on this issue in the discussion:

*"In scenario (ii) cliff retreat rates tended toward a constant rate with the result that the distribution of $^{10}$Be was approximately constant through time (Figure 14). The concentrations of $^{10}$Be were not consistent with a steady state evolution scenario in which the platform gradient is fixed, which predicted roughly twice the amount of $^{10}$Be for a particular position on the platform. The difference can be explained by the dissimilar platform morphology brought about by uneven distribution of platform downwear in the transient model simulations. Greater rates of down-wear in the inter-tidal zone lower the platform more rapidly in the near-shore, which removes $^{10}$Be-laden rock and results in deeper water in the near-shore (and therefore reduced $^{10}$Be production) relative to the steady state model runs that assume constant and uniform downwear. These differences may be exacer-bated when taking into account the spatial distribution of other processes such as weathering, the spatial distribution of which is currently poorly understood (Porter et al., 2010)."*

**References**

Anderson, R.S., Densmore, A.L., and Ellis, M.A., 1999, The generation and degrada-tion of marine terraces: Basin Research, v. 11, no. 1, p. 7–19, doi: 10.1046/j.1365-2117.1999.00085.x.

Chao, R.B., Casais, M.C., Cortizas, A.M., Alberti, A.P., and Trenhaile, A.S., 2003, Evo-lution and inheritance of a rock coast: Western Galicia, Northwestern Spain: Earth Surface Processes and Landforms, v. 28, no. 7, p. 757–775, doi: 10.1002/esp.496.

Choi, K.H., Seong, Y.B., Jung, P.M., and Lee, S.Y., 2012, Using Cosmogenic [10]Be Dating to Unravel the Antiquity of a Rocky Shore Platform on the West Coast of Korea: Journal of Coastal Research, v. 282, no. 3, p. 641–657, doi: 10.2112/JCOASTRES-D-11-00087.1.

Lazarus, E.D., McNamara, D.E., Smith, M.D., Gopalakrishnan, S., and Murray, A.B., 2011, Emergent behavior in a coupled economic and coastline model for beach nourishment: Nonlinear Processes in Geophysics, v. 18, no. 6, p. 989–999, doi: 10.5194/npg-18-989-2011.

Matsumoto, H., Dickson, M.E., and Kench, P.S., 2016, An exploratory numerical model of rocky shore profile evolution: Geomorphology, v. 268, p. 98–109, doi: 10.1016/j.geomorph.2016.05.017.

Porter, N.J., Trenhaile, A.S., Prestanski, K., and Kanyaya, J.I., 2010, Patterns of surface downwearing on shore platforms in eastern Canada: Earth Surface Processes and Landforms, v. 35, no. 15, p. 1793–1810, doi: 10.1002/esp.2018.

Regard, V., Dewez, T., Bourlais, D.L., Anderson, R.S., Duperret, A., Costa, S., Leanni, L., Lasseur, E., Pedoja, K., and Maillet, G.M., 2012, Late Holocene seacliff retreat recorded by [10]Be profiles across a coastal platform: Theory and example from the English Channel: Quaternary Geochronology, v. 11, p. 87–97, doi: 10.1016/j.quageo.2012.02.027.

Robinson, L.A., 1977, Erosive processes on the shore platform of northeast Yorkshire, England: Marine Geology, v. 23, no. 4, p. 339–361, doi: 10.1016/0025-3227(77)90038-X.

Rogers, H.E., Swanson, T.W., and Stone, J.O., 2012, Long-term shoreline retreat rates on Whidbey Island, Washington, USA: Quaternary Research, v. 78, no. 2, p. 315–322, doi: 10.1016/j.yqres.2012.06.001.

Stephenson, W.J., Kirk, R.M., Hemmingsen, S.A., and Hemmingsen, M.A., 2010,

Decadal scale micro erosion rates on shore platforms: Geomorphology, v. 114, no. 1–2, p. 22–29, doi: 10.1016/j.geomorph.2008.10.013.

Stephenson, W.J., Kirk, R.M., Kennedy, D.M., Finlayson, B.L., and Chen, Z., 2012, Long term shore platform surface lowering rates: Revisiting Gill and Lang after 32 years: Marine Geology, v. 299–302, p. 90–95, doi: 10.1016/j.margeo.2012.01.005.

Sunamura, T., 1992, Geomorphology of Rocky Coasts: John Wiley and Sons Ltd.

Trenhaile, A.S., 2014a, Climate change and its impact on rock coasts: Geological Society, London, Memoirs, v. 40, no. 1, p. 7–17, doi: 10.1144/M40.2.

Trenhaile, A.S., 2004, Modeling the accumulation and dynamics of beaches on shore platforms: Marine Geology, v. 206, no. 1–4, p. 55–72, doi: 10.1016/j.margeo.2004.03.013.

Trenhaile, A.S., 2000, Modeling the development of wave-cut shore platforms: Marine Geology, v. 166, no. 1–4, p. 163–178, doi: 10.1016/S0025-3227(00)00013-X.

Trenhaile, A.S., 2001a, Modeling the effect of late Quaternary interglacial sea levels on wave-cut shore platforms: Marine Geology, v. 172, no. 3–4, p. 205–223, doi: 10.1016/S0025-3227(00)00136-5.

Trenhaile, A., 2014b, Modelling the effect of Pliocene-Quaternary changes in sea level on stable and tectonically active land masses: Earth Surface Processes and Landforms, v. 39, no. 9, p. 1221–1235, doi: 10.1002/esp.3574.

Trenhaile, A.S., 2001b, Modelling the quaternary evolution of shore platforms and erosional continental shelves: Earth Surface Processes and Landforms, v. 26, no. 10, p. 1103–1128, doi: 10.1002/esp.255.

Walkden, M.J.A., and Hall, J.W., 2005, A predictive Mesoscale model of the erosion and profile development of soft rock shores: Coastal Engineering, v. 52, no. 6, p. 535–563, doi: 10.1016/j.coastaleng.2005.02.005.

---

## Author Comment (AC3) · 6 Oct 2016

**Response to review**

We are grateful to Dr. Mark Dickson for his positive and constructive comments, and extend our thanks also to Hironori Matsumoto for his contributions to the review. We welcome the inclusion of aspiring PhD students in the peer review process. Below we detail our responses to all of these comments, highlighting any changes we have made to the manuscript guided by the reviewer's suggestions. We have numbered the reviewer comments and they are coloured black, while our responses are coloured blue, and italicised where we are quoting directly from the revised manuscript.

[Figure]

1. This is a welcome addition to a rather slowly emerging literature on the application of cosmogenic dating to inform rocky shore evolution studies. The proposed model usefully extends the work described by Regard et al. (2012) and includes I think the majority of the factors required. In doing so, the paper highlights that the use of this method to determine erosion rates on rocky shores, while conceptually simple, in practice is rather more complex. Nevertheless, the lack of quantitative measurements of rates of change has been a long-standing limitation in our field, so efforts in the direction of this paper are welcome.

   We thank the reviewer for these positive comments and hope this work can be useful for guiding future applications of cosmogenic radionuclides in rock coast settings.

2. Below are some comments that the authors might wish to consider during any revision. Please note that these comments integrate thoughts from Hironori Matsumoto who is currently making further developments to a rocky shore profile evolution model (Matsumoto, H., Dickson, M. E., and Kench, P. S.: An exploratory numerical model of rocky shore profile evolution, Geomorphology, 268, 98–109, doi:10.1016/j.geomorph.2016.05.017, 2016.)

   Thank you for bringing this manuscript to our attention, we had already cited it in the manuscript. We thank Hironori Matsumoto for his contributions to the review, and more generally we welcome the consultation of PhD students in an open review process.

3. Can you clarify how the model considers platform slope? Does it calculate the cliff toe position with a fixed platform geometry, similar to previous tide-less models (Sunamura, 1975)? Perhaps not, because later, in section 5.5, there is reference to the gradient of the shore platform decreasing through time, as the platform widens. There is also reference to the emergence of a 'stepped' platform - what is the reason for this? Ultimately it is a little unclear how exactly the geometry

is handled. The text states that the model is similar to the existing models of Trenhaile (2000), Walkden and hall (2005), and Matsumoto et al (2016), which all consider tide and vertical components. It would be useful to expand and clarify similarities and differences in that regard.

We apologise that the details of the morphological development were unclear and are pleased to have the opportunity to clarify these points. There are two morphological models used in this study. The first is a steady-state model in which a constant shore profile characterised by a fixed platform gradient is translated landward through time, tracking the elevation of mean sea level. Therefore, in these steady-state scenarios, tides do not influence the morphological evolution of the platform. However tides still modify the production of $^{10}$Be on the platform due to water shielding. In the second modelling approach, a dynamic morphological model is used, and it is this model that is broadly similar to Trenhaile (2000) and Matsumoto et al. (2016). The dynamic model calculates the point of wave breaking and therefore the width of the surf zone, which are dependent on the instantaneous mean water level which is set by the tides. Horizontal erosion at the water level is proportional to the delivery of wave energy (equation 1), parameterised as the height of the breaking wave once it has crossed the surf zone (equation 4). The platform is also eroded at depth below the water line, with the amount of erosion declining exponentially with depth. Stepped platforms can develop because the total amount of erosion integrated over a tidal cycle has two maxima as a function of elevation, one just below the upper tidal limit, and one just above the lower tidal limit (similar to Figure 1 in Trenhaile, 2000).

We have restructured the beginning of section 2 to be more explicit about the different approaches taken to morphological modelling:

*"Cliffed, rocky coasts, are commonly fronted by shore platforms that have previously been classified into two types (Sunamura,1992). Type-A platforms are characterised by a gently sloping erosional platform surface extending offshore*

*beyond maximum low water. Type-B platforms are shallow gradient to sub-horizontal and terminate at their seaward edge at maximum low water through a scarp (Figure 2a). Numerical models of shore platform evolution have successfully recreated both of these endmember morphologies, but have revealed that there are a range of other possible morphologies in between, for example sloping platforms terminating in a scarp (e.g. Trenhaile, 2000; Walkden and Hall, 2005; Matsumoto et al., 2016)."*

*"Numerical models of platform evolution demonstrate that shore platforms and adjacent sea cliffs tend towards a morphological steady state, Under such conditions the morphology may reflect the combination of RSL change, tides and wave energy availability. However, the assumption of steady state retreat may not always be applicable (Dickson et al., 2013). We expected the style of platform evolution to be important for the distribution of [10]Be across a shore platform. Moreover, micro-erosion meter measurements of platform downwear suggest that downwear is not uniform across the shore profile, but tends to be faster in the upper inter-tidal zone and decline with depth (Porter et al., 2010), and to explore the influence of such a distribution on [10]Be concentrations, a dynamic morphological model was required."*

*"Therefore, two different morphological models are used in this study. The first assumes steady state evolution of the shore profile, such that coastal morphology does not change its form through time, and a constant cliff-platform geometry is translated landward through time. The second model is a dynamic shore platform evolution model similar to that of Trenhaile (2000) that can reproduce a range of platform geometries. These two approaches are described in more detail in the subsequent sections."*

4. Did the authors consider prospects for using the model to test/discuss factors that affect [10]Be concentrations on other types of shore platform geometry beyond the sloping (type-A) platform investigated? The paper refers to type-B platforms,

raising the question as to whether the model adds any new knowledge of likely differences in concentrations across different possible platform geometries.

*Please see also our response to Alan Trenhaile's similar comment (response #7 in the response to Alan Trenhaile's review). We are not exclusively considering type-A and type-B platforms, these are end-member cases, and the dynamic morphological model used in this study produces alternative platform geometries such as stepped platforms. We now write:*

*"Cliffed, rocky coasts, are commonly fronted by shore platforms that have previously been classified into two types (Sunamura, 1992). Type-A platforms are characterised by a gently sloping erosional platform surface extending offshore beyond maximum low water. Type-B platforms are shallow gradient to sub-horizontal and terminate at their seaward edge at maximum low water through a scarp (Figure 2a). Numerical models of shore platform evolution have successfully recreated both of these end-member morphologies, but have revealed that there are a range of other possible morphologies in between, for example sloping platforms terminating in a scarp (e.g. Trenhaile, 2000; Walkden and Hall, 2005; Matsumoto et al., 2016)."*

*"Therefore, two different morphological models are used in this study. The first assumes steady state evolution of the shore profile, such that coastal morphology does not change its form through time, and a constant cliff-platform geometry is translated landward through time. The second model is a dynamic shore platform evolution model similar to that of Trenhaile (2000) that can reproduce a range of platform geometries. These two approaches are described in more detail in the subsequent sections."*

5. The model considers topographic shielding in the case of a constant cliff height. A further exploration of interest would be to consider a slowly increasing cliff height. What comes to mind is the example of progressive cliffing into hillslopes rounded over long glacial periods. This could introduce further complications for [10]Be con-

centrations, because erosion into cliffs of increasing height might progressively increase beach thickness (5.4.1)... assuming no gradient in alongshore sediment flux. The number of potential model scenarios can quickly increase, but it would be interesting to have a somewhat expanded discussion of this factor.

On the first part of your comment, we agree that it is unlikely on many coast-lines that cliff heights have been constant during platform development but will depend on the morphology of the landscape that is being eroded into. Generally, topographic shielding only has a minor influence on CRN concentrations, since the shielding factors increase rapidly and non-linearly toward unity moving away from the cliff (Figure 4). To illustrate this we have attached a plot showing [10]Be concentrations for a simple case of steady-state platform retreat at 10 cm yr[-1] for a 1/100 gradient platform for cliff heights of 0, 25 and 50 m. The difference in peak concentrations between the 0 and 50m cliff are < 10%. A gradual changing of cliff height would therefore not make a significant difference to the predicted concentrations.

With the second part of your comment we acknowledge that we offered little discussion of the feedbacks between cliffs and beaches. It was intentional at this stage not to perform experiments exploring these feedbacks because as yet they are poorly represented in existing morphological models, particularly when considering alongshore transport.

6. There is some interesting discussion on the implications of platform downwear, but I agree with the other reviewer that this aspect of the modelling could be improved - he makes a number of points to consider in that regard which I have not expanded on here.

We have made a number of changes to the manuscript to address Alan Tren-haile's comments and point the reviewer to these for details (see responses to Alan Trenhaile's review: #3, #5, #6, #8 and #13). We have not changed our mod-elling approach to include more specific processes related to downwear as we

think this would distract from our intention to provide a heuristic exploration of the controls on shore platform [10]Be concentrations.

7. In concluding, is it possible to tabulate or summarise somehow the relative sensitivities of the different factors?

Thank you for this suggestion, we agree that such a summary would be helpful and are considering different option for how best to achieve this in redrafting the manuscript.

8. Please double check that you have the exponent written correctly in equation 5 (i.e. '+' hw(t))?

This should have been negative, thank you for noticing. We have removed the brackets so that the signs are now correct.

9. Please double check the wording of the last sentence on p7 (upper / lower?)

These were correct but we see how it might have been confusing. These changes are relative to a scenario with no tides at all. The upper platform experiences periodic submergence and emergence during a tidal cycle, but the net effect is a reduction in the overall [10]Be production, whilst on the lower platform the net effect is an increase in the overall [10]Be production. We have rewritten this section to clarify:

*"Tides modify production in the platform by varying water depth hw and intermittently submerging and exposing the platform sub-aerially. Relative to a scenario with no tidal variation, Regard et al. (2012) demonstrated that tides have a net effect to reduce [10]Be production in the upper inter-tidal platform due to periodic platform submergence that reduces the net cosmic ray flux received, while [10]Be production in the lower platform increases due to periodic exposure."*

10. two "and"s p4 line 28 - Clarify what is meant by significant (para 8, line 16)

Corrected

11. Mis-spelling of fund[e]mentally

Corrected

12. Grammar problem para 9 line 24

We were unable to find this error. We contacted the reviewer directly who was not able to clarify an error. If this mistake exists and has not been corrected by our redrafting and proof reading, we hope it will be caught during typesetting.

13. no full stop before Failure p15 l12

Corrected

14. Choi2012 p16

Corrected

**References**

Dickson, M.E., Ogawa, H., Kench, P.S., and Hutchinson, A., 2013, Sea-cliff retreat and shore platform widening: Steady-state equilibrium? Earth Surface Processes and Landforms, v. 38, no. 9, p. 1046–1048, doi: 10.1002/esp.3422.

Matsumoto, H., Dickson, M.E., and Kench, P.S., 2016, An exploratory numerical model of rocky shore profile evolution: Geomorphology, v. 268, p. 98–109, doi: 10.1016/j.geomorph.2016.05.017.

Porter, N.J., Trenhaile, A.S., Prestanski, K., and Kanyaya, J.I., 2010, Patterns of surface downwearing on shore platforms in eastern Canada: Earth Surface Processes and Landforms, v. 35, no. 15, p. 1793–1810, doi: 10.1002/esp.2018.

[Figure]

Regard, V., Dewez, T., Bourlais, D.L., Anderson, R.S., Duperret, A., Costa, S., Leanni, L., Lasseur, E., Pedoja, K., and Maillet, G.M., 2012, Late Holocene sea-cliff retreat recorded by [10]Be profiles across a coastal platform: Theory and example from the English Channel: Quaternary Geochronology, v. 11, p. 87–97, doi: 10.1016/j.quageo.2012.02.027.

Sunamura, T., 1975, A Laboratory Study of Wave-Cut Platform Formation: The Journal of Geology, v. 83, no. 3, p. 389–397, doi: 10.1086/628101.

Sunamura, T., 1992, Geomorphology of Rocky Coasts: John Wiley and Sons Ltd.

Trenhaile, A.S., 2000, Modeling the development of wave-cut shore platforms: Marine Geology, v. 166, no. 1–4, p. 163–178, doi: 10.1016/S0025-3227(00)00013-X. Walkden, M.J.A., and Hall, J.W., 2005, A predictive Mesoscale model of the erosion and profile development of soft rock shores: Coastal Engineering, v. 52, no. 6, p. 535–563, doi: 10.1016/j.coastaleng.2005.02.005.

**ESurfD**